# Predictive Coding Beyond Gaussian Distributions

**Luca Pinchetti**[1] **Tommaso Salvatori**[1] **Yordan Yordanov**[1]
**Beren Millidge**[2] **Yuhang Song**[1,2,†] **Thomas Lukasiewicz**[3,1]
[1] Department of Computer Science, University of Oxford, UK
[2] MRC Brain Network Dynamics Unit, University of Oxford, UK
[3] Institute of Logic and Computation, TU Wien, Austria
luca.pinchetti@cs.ox.ac.uk, tommaso.salvatori@cs.ox.ac.uk
yordan.yordanov@cs.ox.ac.uk, beren.millidge@ndcn.ox.ac.u
yuhang.song@some.ox.ac.uk, thomas.lukasiewicz@cs.ox.ac.uk

## Abstract

A large amount of recent research has the far-reaching goal of finding training methods for deep neural networks that can serve as alternatives to backpropagation (BP). A prominent example is predictive coding (PC), which is a neuroscience-inspired method that performs inference on hierarchical Gaussian generative models. These methods, however, fail to keep up with modern neural networks, as they are unable to replicate the dynamics of complex layers and activation functions. In this work, we solve this problem by generalizing PC to arbitrary probability distributions, enabling the training of architectures, such as transformers, that are hard to approximate with only Gaussian assumptions. We perform three experimental analyses. First, we study the gap between our method and the standard formulation of PC on multiple toy examples. Second, we test the reconstruction quality on variational autoencoders, where our method reaches the same reconstruction quality as BP. Third, we show that our method allows us to train transformer networks and achieve a performance comparable with BP on conditional language models. More broadly, this method allows neuroscience-inspired learning to be applied to multiple domains, since the internal distributions can be flexibly adapted to the data, tasks, and architectures used.

## 1 Introduction

The last decade has seen an explosion of machine learning research fueled by the collection of an unprecedented amount of data and the development of models that can make use of it. Starting with AlexNet [Krizhevsky et al., 2012], deep neural networks trained with *backpropagation (BP)* [Rumelhart et al., 1986] have been established as the best-performing models in many fields [Silver et al., 2016, He et al., 2016, Brown et al., 2020, Ramesh et al., 2021, Devlin et al., 2018]. Despite reaching human-level performance in several tasks [Silver et al., 2016, Vinyals et al., 2017, 2019], we are still far from artificial general intelligence. The trend has been to constantly increase the number of parameters in such networks, from millions [Devlin et al., 2018] to hundreds of billions [Brown et al., 2020]. The limitations and drawbacks given by the large size of modern architectures have motivated research that looks for alternative methods to train them. The direction of research that has inspired this work is that of neuroscience-inspired alternatives to BP, which promise to overcome these drawbacks, due to both interesting properties of their credit assignment, such as plasticity [Hebb, 1949], and their biological hardware [Kendall et al., 2020]. These methods have two main advantages relative to standard deep learning models trained with BP. First, it is much more feasible to train them on analog and neuromorphic chips [Kendall et al., 2020]. Second, the resulting

---

† Corresponding author.

computational models are extremely flexible in both network layout design and querying techniques [Salvatori et al., 2022a, Millidge et al., 2021, Salvatori et al., 2021, Papadimitriou et al., 2020]. These two properties could play a crucial role in overcoming the limitations of BP-based learning towards artificial general intelligence.

*Predictive coding (PC)* is one of the most influential theories of information processing in the brain, initially proposed to explain a large number of brain behaviours [Mumford, 1992, Friston and Kiebel, 2009], and now also a topic of research in machine learning, thanks to the computational model proposed by Rao and Ballard [1999]. This method has in fact been used in supervised and unsupervised tasks [Ororbia and Kifer, 2020, Whittington and Bogacz, 2017, Han et al., 2018], with an important theoretical result: the original formulation is able to approximate the weight update of BP [Whittington and Bogacz, 2017, Millidge et al., 2020], and to exactly replicate it when introducing small variations [Salvatori et al., 2022b, Song et al., 2020]. These results are important, as they draw a strong connection with the aforementioned results obtained by BP in the last decade. PC, however, also presents several interesting properties that make it different from BP: it has an energy-based formulation that allows the design of powerful associative memory models [Salvatori et al., 2021] and to train graphs of any topology [Salvatori et al., 2022a].

PC can also be studied from an information theory perspective, as it is a hierarchical generative model [Rao and Ballard, 1999]. This is a strength of the model, as it has allowed such models to achieve a competitive performance to standard models trained with BP on several generative tasks. These results, however, are all obtained on simple models: sequential architectures with element-wise activations and quadratic energy functions. Deep learning, however, has progressed far from those in recent years, and, therefore, it is necessary to evaluate the performance of PC on more up-to-date and complex architectures to obtain a complete comparison of PC against BP. In this paper, we see that the strict Gaussian assumption is limiting when dealing with more complex architectures such as transformers, preventing PC from reaching the performance obtained by BP. Note that this limitation is not unique to PC, but it is shared among all neuroscience-inspired methods: for example, to our knowledge, none of these frameworks has been successfully used to train language models to date.

In this work, we address this problem by generalizing PC to arbitrary distributions. This allows us to both use Gaussian distributions when allowed, and also to deal with intractable ones by approximating them using a sampling scheme. The resulting framework is coherent with the PC theory, as it enables the definition of layer-wise energy functions that represent prediction errors [Rao and Ballard, 1999, Whittington and Bogacz, 2017]. In standard PC networks, the error is given by the difference between the expected and actual input of a layer; here, it is defined as the KL-divergence between the expected and actual distributions. We show that these formulations are equivalent when using Gaussian distributions, meaning that our proposed framework is a generalization of standard PC. The results of this paper are briefly summarized as follows:

- We generalize PC beyond the assumption of a Gaussian generative model. This lets us define prediction errors as "distances" between arbitrary distributions at each hierarchical layer of a PC network, and derive a novel free-energy objective to train such networks.

- We empirically show that standard PC is ineffective in training models with complex structure and activation functions, as they cannot be approximated by a Gaussian generative model. Our proposed method, instead, significantly outperforms it, reaching a competitive performance with BP in training both variational autoencoders [Kingma and Welling, 2014] and transformers [Vaswani et al., 2017]. This further bridges the gap in performance between state-of-the-art deep learning methods and neuroscience-inspired learning.

The rest of this paper is structured as follows. In Section 2, we introduce the notation used throughout the paper and describe the probabilistic interpretation of PC. In Section 3, we propose how to generalize the definition of PC beyond Gaussian generative models. In Section 4, we evaluate the performance of our proposed method in comparison to standard PC and BP. Finally, in Sections 5 and 6, we discuss related work and provide a conclusion, respectively.

## 2 Predictive Coding: A Probabilistic Interpretation

In this work, we focus on multi-layer networks organised in a sequence of $L$ layers. These networks define a model $\mathcal{M}_\theta$ that learns the relationship between the input $d$ and the target $o$ by updating

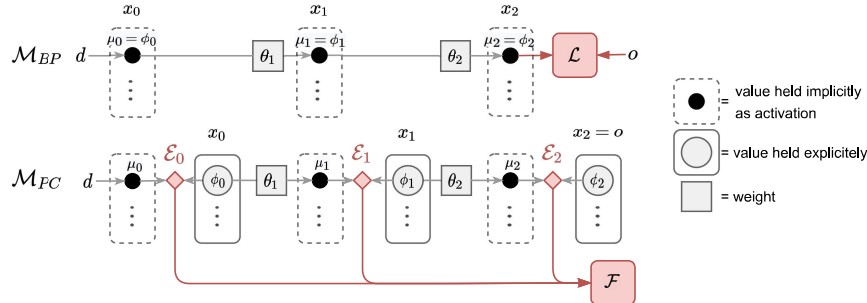

Figure 1: Difference between a network trained with BP (up) and PC (down). The nodes $x_l$ of each layer store the extra parameters $\phi_l$. By following the computational arrows backward, the error flows globally from the last layer to the first when using BP. With PC, instead, each layer computes a local error that gets propagated only to nearby nodes.

its weight parameters $\theta = (\theta_1, \ldots, \theta_L)$. Networks trained with BP have a forward pass, where an output $\hat{o}$ is computed from an input $d$, and a backward pass, where the error given by a loss function $\mathcal{L} = loss(o, \hat{o})$ is propagated back to the network. Every layer of the model has a set of value nodes $x_l$, which contain the value of the prediction $\mu_l$ (i.e., what is normally called activation of a layer), computed accordingly to the value of the previous layer. In detail, we have $\mu_l = f_l(x_{l-1}, \theta_l)$, where $f_l$ is an activation function. In PC, the prediction $\mu_l$ is not stored directly inside the value nodes $x_l$, but rather in separate prediction nodes. The nodes $x_l$ hold, instead, the neural activation $\phi_l$ of layer $l$. $\phi_l$ is a parameter of the network that can be optimized by minimizing the so-called prediction error between $\mu_l$ and itself. Therefore, a model trained via BP consists exclusively of the parameters $\theta$, while a PC network requires two sets of parameters, $\theta$ and $\phi$, where $\phi = (\phi_0, \ldots, \phi_L)$ represents the neural activities of different layers (Fig. 1). Table 1 summarises the differences between BP and PC.

| Method | Value received from $l-1$ | Content of value nodes $x_l$ | Value fed to layer $l+1$ |
|--------|---------------------------|------------------------------|--------------------------|
| BP | $\mu_l$ | prediction from $l-1$ ($\mu_l$) | $\mu_l$ |
| PC | $\mu_l$ | neural activation of $l$ ($\phi_l$) | $\phi_l$ |

Table 1: Clarification of the notation used, highlighting the differences between BP and PC.

## 2.1 PC as Variational Inference

PC as a learning algorithm can be mathematically interpreted as a variational inference problem. It is in fact possible to consider learning as an intractable Bayesian inference process that can be approximated with a tractable optimization problem [Friston, 2003, 2005, 2008]. Under this assumption, the neural activities $\phi_i$ in a PC network represent probability distributions. In particular, Friston based his theory of PC on Gaussian generative models. A detailed review is provided in [Millidge et al., 2021]. Assume that we have a generative model $o = g(x)$, where $o$ is a data point and $x$ a set of latent variables, which is described by the joint probability $p(o, x) = p(o|x)p(x)$. We need to solve an inverse problem: given a data point $o$, we need to infer the causes $x$ that generate $o$ through $g$. Similarly to many inverse problems, this one is intractable. In particular, we want to compute the true posterior $p(x|o)$. However, computing it by Bayes rule as $p(x|o) = p(o, x)/p(o)$ requires the normalizing factor $p(o) = \int p(x, o)dx$, which is, for all but trivial problems, intractable. Variational inference aims to approximate the intractable posterior with a family of distributions $q_\phi(x|o)$, where the parameters $\phi$ have to be learnt. This is generally done via gradient descent on the KL divergence [Kullback and Leibler, 1951] between the approximated and true posterior. The goal is to compute

$$q_\phi^* = \underset{\phi}{\operatorname{argmin}} \, D_{KL}[q_\phi(x|o)||p(x|o)] \tag{1}$$

by minimizing an upper bound on the divergence, called the variational free energy $\mathcal{F}$:

$$\mathcal{F} := D_{KL}[q_\phi(x|o)||p(o, x)] \geq D_{KL}[q_\phi(x|o)||p(o, x)] + \ln p(o) = D_{KL}[q_\phi(x|o)||p(x|o)]. \tag{2}$$

The PC framework assumes a Gaussian form for the generative model $p(o, x) = p(o|x)p(x) = \mathcal{N}(o; f(x, \theta), \Sigma_2)\,\mathcal{N}(x, \mu, \Sigma_1)$, where $\Sigma_2$, $\Sigma_1$, and $\mu$ are prior parameters that can optionally be learnt. Using as variational posterior the Dirac-delta distribution* $q_\phi(x|o) = \delta(x - \phi)$, we get that

$$\mathcal{F} = \mathbb{E}_{q_\phi(x|o)}[\ln q_\phi(x|o)] - \mathbb{E}_{q_\phi(x|o)}[\ln p(o, x)] = -\mathbb{E}_{q_\phi(x|o)}[\ln p(o, x)] = -\ln p(o, \phi), \quad (3)$$

where the entropy of $q$ is 0. This scheme can be applied to deep neural networks, where $x$ does not represent a homogeneous latent space (e.g., a single layer), but is, instead, organised in a hierarchical structure, defined by the multiple layers of a PC network with widths $w_1, \ldots, w_l$ and nodes $x_0, x_1, \ldots, x_L$. Under this premise, the generative model is as follows:

$$p(x_{0:L}) = p(x_0) \prod_{l=1}^{L} p(x_l|x_{l-1}) = \mathcal{N}(x_0; \mu_0, \Sigma_0) \prod_{l=1}^{L} \mathcal{N}(x_l; \mu_l, \Sigma_l), \quad (4)$$

where $\mu_l = f_l(x_{l-1}, \theta_l)$, and $x_L$ corresponds to the observation layer and is set to $x_L = o$ during training. The parameters $\Sigma_l$ are prior diagonal covariance matrices, which can be optionally learnt, and $\mu_0$ is an arbitrary prior that can be set to some given data $d$. This is equivalent to the training of a supervised network with data point $d$ and label $o$. The energy becomes:

$$\widetilde{\mathcal{F}} = -\mathbb{E}_{q_\phi(x_{0:L}|d,o)}[\ln p(x_{0:L})] = \sum_{l=0}^{L} -\ln p(\phi_l|\mu_l) = \frac{1}{2}\Big(\sum_{l=0}^{L} \sum_{i=1}^{w_l} \Sigma_{l,i}^{-1} \epsilon_{l,i}^2 + \ln \Sigma_{l,i}\Big) + k, \quad (5)$$

where $k$ is a constant, $\epsilon_l = (\phi_l - \mu_l)$, and $q_\phi(x_{0:L}|d,o) = \prod_{l=0}^{L} \delta(x_l - \phi_l)$ (implying that $x_l = \phi_l$). In this equation, the total energy is given by the sum of the energies $\mathcal{E}_l$ of every layer, where $\mathcal{E}_l := -\ln p(\phi_l|\mu_l)$. By assuming identity covariance matrices (i.e., $\Sigma_l = I$)†, the energy becomes the sum of quadratic errors, introduced by Rao and Ballard [1999]:

$$\mathcal{F} = \sum_{l=0}^{L} \mathcal{E}_l = \sum_{l=0}^{L} \epsilon_l^2. \quad (6)$$

In most cases, however, the generative model depends on a set of parameters $\theta$: $p(x_0, \ldots, x_L; \theta)$ that need to be learned according to a specific dataset. This can be done via expectation maximization (EM) [Dempster et al., 1977], where we first infer the best possible latent variables $\phi$ given a data point $o$ (E-step) and then use them to update the parameters $\theta$ (M-step). In practice, both these phases are achieved using gradient descent to minimize $\mathcal{F}$. In detail, given a labelled point $(d, o)$, the input layer prior is set to $\mu_0 = d$, while the output layer nodes are fixed to $\phi_L = o$ for both the inference phase (E-step) and weight update (M-step). During the inference phase, the weight parameters are fixed, and the node values $\phi$ are continuously updated via gradient descent to minimize the energy $\mathcal{F}$. This process either runs until convergence or for a fixed number of steps $T$. When the inference phase ends, a weight update is performed as follows: the node values $\phi$ are fixed, and the weights are updated once via gradient descent on the same energy function $\mathcal{F}$.

## 3  Generalization to Arbitrary Distributions

In this section, we go beyond the strict Gaussian assumption of PC. According to Eq. 5, we have that $\mathcal{E}_l = -\ln p(\phi_l|\mu_l) = -\ln \mathcal{N}(\phi_l; f_l(\phi_{l-1}, \theta_l), \Sigma_l))$. We can highlight the role of each $\phi_i$, by introducing the $\cdot^{\mathcal{D}}$ and $\cdot^{\mathcal{S}}$ superscripts, which, respectively, indicate that a vector value is interpreted as a distribution (i.e., a vector of sufficient statistics that uniquely identifies a probability distribution, such as the mean $u$ of a Gaussian distribution), or as a single sample. The $\cdot^{\mathcal{S}}$, $\cdot^{\mathcal{D}}$ notation does not, in any case, imply any transformation of the value itself. We get that

$$\mathcal{E}_l = -\ln \mathcal{N}(\phi_l^{\mathcal{S}}; f_l(\phi_{l-1}^{\mathcal{D}}, \theta_l), \Sigma_l)) = -\ln p(\phi_l^{\mathcal{S}}|\phi_{l-1}^{\mathcal{D}}, \theta_l, \Sigma_l). \quad (7)$$

Thus, neural activation $\phi_l$ is simultaneously interpreted both as $\phi_l^{\mathcal{S}}$ and $\phi_l^{\mathcal{D}}$. This subtle difference has never been highlighted in standard works using hierarchical Gaussian models, since $\phi_l^{\mathcal{S}}$ corresponds to the maximum likelihood estimation of the Gaussian distribution $\mathcal{N}(\phi_l^{\mathcal{D}}, \Sigma_l)$, and thus $\phi_l^{\mathcal{S}} = \phi_l^{\mathcal{D}}$.

---

*A Gaussian variational posterior under the Laplace approximation can also be used, resulting in the same learning rules as the PC framework proposed here; see [Buckley et al., 2017].

†Throughout the paper, we assume diagonal covariance matrices for the Gaussian generative model in order to simplify the mathematical derivations. However, our approach can be naturally extended to the case of general covariance matrices [Bogacz, 2017].

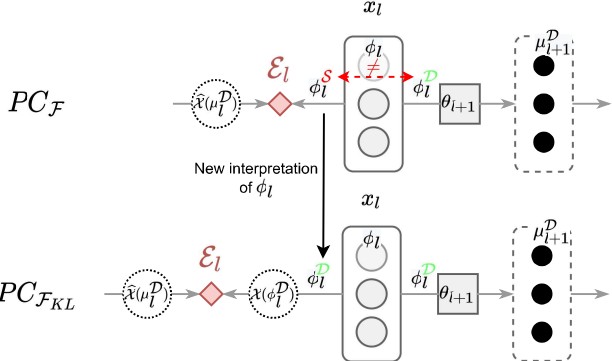

Figure 2: Different layer structure between $PC_{\mathcal{F}}$ (left) and $PC_{\mathcal{F}_{KL}}$ (right). In $PC_{\mathcal{F}}$, the nodes of each layer are simultaneously interpreted as both samples (when evaluating $\mathcal{E}_l$) and as distributions (when computing $\mu_{l+1}$). This inconsistency (highlighted by the dashed red arrow) is not present in $PC_{\mathcal{F}_{KL}}$, where they always represent probability distributions.

Assuming a Gaussian form for the generative model could, however, become a limiting factor when considering complex architectures. For example, a layer employing a *softmax* activation cannot be easily approximated through a multi-variate Gaussian, since the function introduces a strong dependency between the nodes (i.e., their values have to sum to 1). As we will show in the next sections, using *softmax* activation functions can hinder the ability of a network to learn using the standard definition of PC, and hence a generalized formulation that goes beyond the standard Gaussian assumption is needed.

To generalize and extend the canonical formulation of PC, we do the following: instead of directly maximizing the likelihoods $p_l := p(\phi_l^{\mathcal{S}}|\mu_l^{\mathcal{D}})$, we assume that this process is happening repeatedly between each pair of consecutive layers for a single optimization step. We consider the nodes $x_l$ as storing the sufficient statistics $\phi_l^{\mathcal{D}}$ of an arbitrary distribution $\mathcal{X}_l(\phi_l^{\mathcal{D}})$, [‡] we draw $N$ sample points from it, $s_l^{(i)} \sim \mathcal{X}_l(\phi_l^{\mathcal{D}})$, $i \in \{1, \dots, N\}$, and we minimize each individual likelihood $p_l^{(i)} = p(s_l^{(i)}|\mu_l^{\mathcal{D}})$. Furthermore, we remove the Gaussian assumption and consider $\mu_l^{\mathcal{D}} = f_l(\phi_{l-1}^{D}, \theta_l)$ to be a parametrization of a generic distribution $\widehat{\mathcal{X}}_l(\mu_l^{\mathcal{D}})$. By doing so, the node values $\phi_l$ are interpreted exclusively as distribution parameters: both the energies $\mathcal{E}_l$ and the activations $\mu_l^{\mathcal{D}}$ are functions of $\phi_l^{\mathcal{D}}$. It follows that the variational free energy $\mathcal{F}$ is also a function of the expected values given by the likelihoods $p(s_l^{(i)}|\mu_l^{\mathcal{D}})$, $i \in \{1, \dots, N\}$, for each layer $l$. The energy of each layer is then defined as:

$$\bar{\mathcal{E}}_l := -\ln p(\phi_l^{\mathcal{D}}|\mu_l^{\mathcal{D}}) \approx \mathcal{H}(\mathcal{X}_l(\phi_l^{\mathcal{D}}), \widehat{\mathcal{X}}_l(\mu_l^{\mathcal{D}})). \tag{8}$$

A detailed derivation of the above equation is presented in the supplementary material. Knowing that the cross-entropy between two distributions $\mathcal{H}(a, b) = D_{KL}[a||b] + \mathcal{H}(a)$ and that $\mathcal{H}(a) \geq 0$, the total energy of the network can be optimized by minimizing

$$\mathcal{F}_{KL} = \sum_{l=0}^{L} \mathcal{E}_l := \sum_{l=0}^{L} D_{KL}[\mathcal{X}_l(\phi_l^{\mathcal{D}})||\widehat{\mathcal{X}}_l(\mu_l^{\mathcal{D}})] \leq \sum_{l=0}^{L} \mathcal{H}(\mathcal{X}_l(\phi_l^{\mathcal{D}}), \widehat{\mathcal{X}}_l(\mu_l^{\mathcal{D}})). \tag{9}$$

This follows, as the entropy of $\mathcal{X}_l(\phi_l^{\mathcal{D}})$ can be assumed as being a constant depending on the training data. Figure 2 highlights the difference between the original and new PC formulation. This definition of the variational free energy $\mathcal{F}_{KL}$ does not rely on the Gaussian generative assumption, and (as long as the KL divergence between $\mathcal{X}_l$ and $\widehat{\mathcal{X}}_l$ can be efficiently computed) there is no limit on the kind of distribution that can be used. Throughout our experiments, we assumed that the distributions $\mathcal{X}_l$ and $\widehat{\mathcal{X}}_l$ belong to the same family, but different families can be used in different layers. In the supplementary material, we show how $\mathcal{F}_{KL}$ is equivalent to $\mathcal{F}$ when assuming a Gaussian generative model. We also analyze the learning dynamics defined by applying the expectation-maximization (EM) algorithm to Eq. (9).

---

[‡] We use the notation $\widehat{\mathcal{X}}_l(\mu_l)$ to emphasize the dependency of the distribution $\widehat{\mathcal{X}}_l$ exclusively on the parameters $\mu_l$. For example, $\widehat{\mathcal{X}}_l$ could be a Gaussian distribution, and $\mu_l$ represents its mean and variance.

# 4 Experiments

We compare the results of different models trained with BP and PC on various tasks. The main goal is to make PC competitive with BP for complex deep neural architectures. Only recently, PC has begun to be applied to train neural networks for classification tasks [Whittington and Bogacz, 2017], with similar performance but a greater versatility compared to BP [Salvatori et al., 2022a]. We show that our extended probabilistic formulation can be applied to more complex architectures. We refer to each specific version of PC by specifying its energy function.

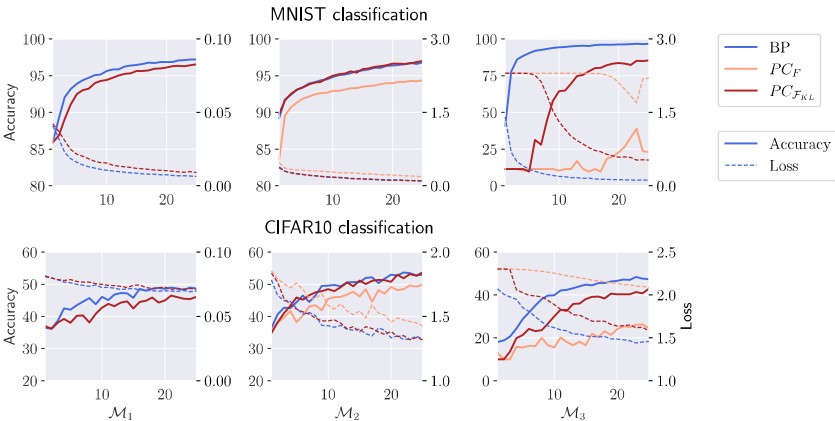

Figure 3: Classification performance of the three models on the MNIST and CIFAR10 datasets. $PC_{\mathcal{F}_{KL}}$ noticeably outperforms $PC_{\mathcal{F}}$, reaching performances comparable with BP. This is true, especially for $\mathcal{M}_2$, which reflects the most commonly used architecture among the three. The x-axis represents the number of epochs.

## 4.1 Classification

As a proof-of-concept experiment, we have trained different models on classification tasks for both the MNIST [Deng, 2012] and CIFAR10 [Krizhevsky et al.] datasets. We evaluated the final accuracy of the model when training with BP compared to PC, as well as the training efficiency, measured as improvements over epochs.

**Setup:** We defined three variations of a fully connected network with $L = 3$ hidden layers and width $w = 512$:

- $\mathcal{M}_1$ uses *tanh* as activation function for the hidden and final layers. The mean squared error (MSE) is the loss function used for BP and to compute the test loss of both PC and BP.

- $\mathcal{M}_2$ uses the *softmax* activation function for the final layer. Consequently, the loss function used is cross-entropy (CE). This architecture corresponds to the one normally used in classification tasks.

- $\mathcal{M}_3$ is a copy of $\mathcal{M}_2$ where the activation function of the second hidden layer is replaced with *softmax*. CE is again the loss function used.

Note that the experiments performed with a *softmax* activation in a hidden layer are merely presented with the goal of empirically validating the effectiveness of our theory, as networks of this kind have never been used in practical tasks. Effectively, $\mathcal{M}_2$ represents the only widely used architecture. We used a weight learning rate of $\beta_\theta = 0.0001$ for both PC and BP. For PC, we used $T = 32$ $\phi$-steps. We assumed identity covariance matrices for the Gaussian distributions of the generative model. Consequently, the energies $\mathcal{F}$ and $\mathcal{F}_{KL}$ differ only for the function used for a *softmax*-activated layer. For $\mathcal{F}_{KL}$, that is

$$\mathcal{E}_{l_{\text{softmax}}} = D_{KL}[\mathcal{X}_l(\phi_l^{\mathcal{D}})||\widehat{\mathcal{X}}_l(\mu_l^{\mathcal{D}})] = \sum_{i=1}^{w_l}(\phi_{l,i}) \cdot \ln(\frac{\phi_{l,i}}{\mu_{l,i}}), \tag{10}$$

where $\mathcal{X}_l$ and $\widehat{\mathcal{X}}_l$ are discrete distributions. Therefore, in the model $\mathcal{M}_1$, $PC_{\mathcal{F}_{KL}}$ and $PC_{\mathcal{F}}$ are algorithmically equivalent (see the supplementary material). More details about the hyperparameters that we used are given in the supplementary material.

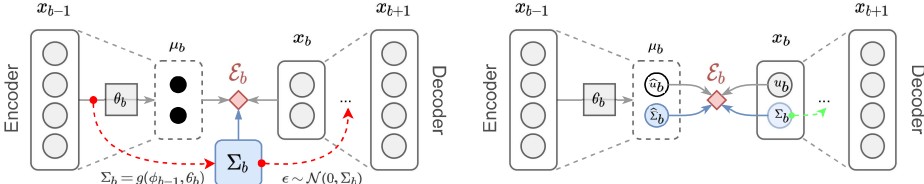

Figure 4: In $PC_{\widetilde{\mathcal{F}}}$ (left), $\Sigma_b$ is stored as an optionally trainable parameter and does not depend on the input $d$. If we were to allow it, and use $\Sigma_b$ to generate $\mu_{b+1}$ (red dashed arrows), we would violate the PC locality assumption, as the error coming from the decoder would flow through $\Sigma_b$ back to the encoder. Using $PC_{\mathcal{F}_{KL}}$ (right), instead, it is possible to have such a dependency by modelling both $\Sigma_b$ and $\widehat{\Sigma}_b$.

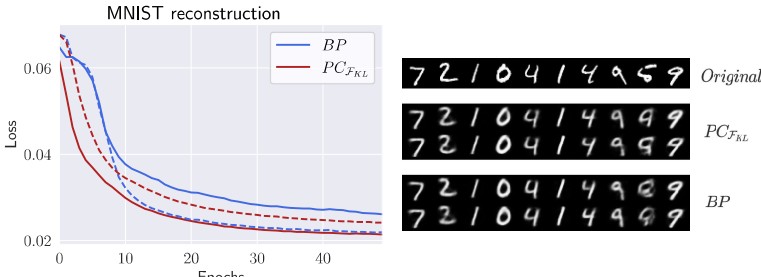

Figure 5: Comparison of BP and PC in training a VAE on the MNIST dataset. The graph on the left shows the test loss over the epochs. The solid and dotted lines represent two different models. Overall PC and BP perform similarly.

**Results:** The results are plotted in Fig. 3. They show that neural networks trained with BP and $PC_{\mathcal{F}_{KL}}$ perform similarly in both $\mathcal{M}_1$ and $\mathcal{M}_2$. The original formulation of PC, on the other hand, performs much worse in all the considered experiments. The first column of plots shows that PC and BP perform similarly when trained with fully Gaussian assumptions. In the experiment on $\mathcal{M}_2$, the standard definition of PC is performing well, even though it is significantly outperformed by BP and $PC_{\mathcal{F}_{KL}}$. When training on $\mathcal{M}_3$, however, the performances are poor. This again does not happen when training using $PC_{\mathcal{F}_{KL}}$, which obtains performances comparable to those of BP. Overall, this clearly indicates that $PC_{\mathcal{F}}$ is not suitable to train neural networks that do not rely uniquely on Gaussian generative models. Instead, we experienced a solid improvement when using the $\widetilde{\mathcal{F}}_{KL}$ energy function. It is enough to reach the same accuracy-over-epochs ratio achieved by BP when using $\mathcal{M}_2$. Under more uncommon architectural choices, such as $\mathcal{M}_3$, the ratio is slightly worse in favour of BP, but still decisively better than $PC_{\mathcal{F}}$. The difference is particularly noticeable in the first epochs. We believe that it may be due to a not ideal initialization of the weights for the PC network, which is currently using the default initialization designed for BP networks [He et al., 2015]. Further research in this direction could improve the training performance.

### 4.2 Variational Autoencoders

Variational autoencoders (VAEs) [Kingma and Welling, 2014] are models that rely on distributions different from Gaussians with fixed covariance matrix. This follows, as the bottleneck layer $b$ of a VAE is required to model the distributions of both the mean and the variance of the latent posterior given a sample $d$, $p(u_b, \Sigma_b|d)$. However, both $PC_{\mathcal{F}}$ and $PC_{\widetilde{\mathcal{F}}}$ are not suitable for that, as they require each layer to represent exclusively the Gaussian mean $u_l$. The optionally learnable parameters $\Sigma_l$ do not depend on the particular input sample $d$. Our proposed method, however, overcomes this limitation by learning the full posterior distribution $\mathcal{N}_d(u_b, \Sigma_b)$. This is done by considering the bottleneck layer $b$ as storing the distribution parameters $\phi_b^{\mathcal{D}} = (u_b, \Sigma_b)$. In this case, $\mu_b^{\mathcal{D}} = (\widehat{u}_b, \widehat{\Sigma}_b) = f_b(\phi_{b-1}^{\mathcal{D}}, \theta_b)$. We then employ the reparametrization trick [Kingma and Welling, 2014] by sampling some Gaussian noise $\bar{\epsilon} \sim \mathcal{N}(0, 1)$ to compute $\mu_{b+1} = f_{b+1}(u_b + \bar{\epsilon}Diag(\Sigma_b^{1/2}), \theta_{b+1})$, which is fed to the next layer. More details are shown in Figure 4.

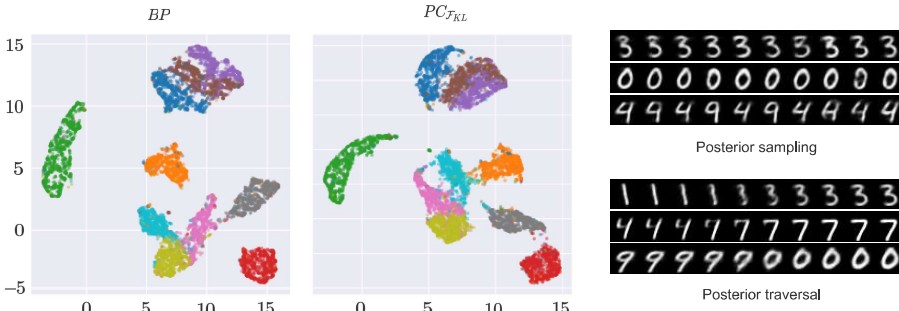

Figure 6: The analysis of the latent space does not highlight any significant differences between PC and BP. UMAP was the algorithm used to obtain the 2D projection [McInnes et al., 2018]. Sampling from the posterior (of the PC-trained model) does not show any anomalies as well.

**Setup:** We trained multiple VAEs on the MNIST dataset, comparing the PC and BP training algorithms. Our architectures employed fully connected layers, with *Hardtanh* activation function for the hidden layers and *sigmoid* for the output layer. The bottleneck layer has a total of $w_b = 32 \, (= 16 + 16)$ latent units. When training with PC, we assumed Gaussian distributions with identity covariance matrix for all but the bottleneck layer, for which we model both the mean $u_b$ and the diagonal covariance matrix $\Sigma_b$ as explained above. We used $T = 32$, but for each data point we sampled a single $\bar{\epsilon}$ at $t = 0$. We use the same weight learning rate $\beta_\theta = 0.0001$ for both BP and PC. Further details about hyperparameters and implementation details are described in the supplementary material.

**Results:** We observed similar results on a wide range of hyperparameters. In Fig. 5, we report the performance of both training methods on two different architectures. The final test loss is overall similar, with neither method being decisively better than the other. The learning curves are also comparable, despite PC being generally faster than BP in the first training epochs. By reconstructing the maximum likelihood estimation of some data points, we can observe how all models produce very similar images. We also performed an analysis of the latent space produced by the encoders and did not detect any significant difference between the two training modes. Figure 6 reports the results. We sampled the latent posterior distribution by encoding a data point $d$ and decoding multiple data points $d'$ obtained by sampling from $\mathcal{N}_d(\mu_b, \Sigma_b)$. To perform a latent traversal, we encoded two different data points, $d_1$ and $d_2$, and reconstructed the maximum likelihood estimation of the vectors obtained by interpolating the two latent embeddings.

### 4.3 Transformer Language Models

To show the performance of our method on more complex tasks, we have trained transformer conditional language models based on the BERT architecture [Devlin et al., 2018]. The conditional language model objective is enforced by modifying the self-attention mechanism with a triangular mask so that no position can attend to any later position in the sequence.

**Setup:** We use the 1B Word Benchmark dataset [Chelba et al., 2013], from which we randomly sample 200,000 training and 10,000 dev instances. We choose to restrict the model's input length to 32 input tokens from a vocabulary of 8001 tokens generated by the BPE-based SentencePiece tokenizer [Kudo and Richardson, 2018]. We use one transformer block with one head and a hidden size of 128 throughout the model. For the PC networks, we assume Gaussian distributions with identity covariance matrix for all but the layers that employ a *softmax* activation function (i.e., the attention layers [Vaswani et al., 2017]). In the latter case, we assume a categorical distribution for the generative model. Consequently, the energy function for those layers is the one defined in Eq. (10). More implementation details and the hyperparameters are given in the supplementary material.

**Results:** For each model, we compare the three training methods $BP$, $PC_\mathcal{F}$, and $PC_{\mathcal{F}_{KL}}$. We found it beneficial to run multiple weight updates for a single training iteration when using $PC_{\mathcal{F}_{KL}}$, but not for $PC_\mathcal{F}$, where it led to instability. We run a hyperparameter search for each training method, select the best models, compare their training curves, and their test performance, and show qualitative examples of model predictions. Figure 7 shows that $PC_{\mathcal{F}_{KL}}$ significantly outperforms $PC_\mathcal{F}$ in

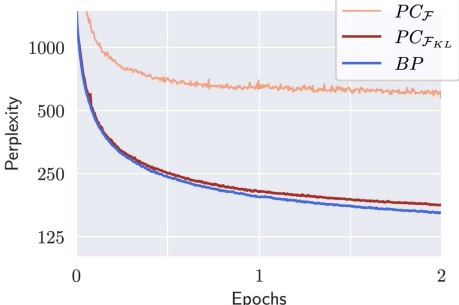

| Training Method | Test Perplexity |
|---|---|
| $BP$ | $162.64 \pm 0.76$ |
| $PC_{\mathcal{F}_{KL}}$ | $175.90 \pm 1.74$ |
| $PC_{\mathcal{F}}$ | $590.08 \pm 12.60$ |

Figure 7: Left: Comparison of language models trained with $BP$ and $PC$, as shown by dev perplexity. Right: Test perplexity achieved by the various training methods for transformer language models. Average ($\pm \sigma$) of 10 seeds.

terms of test perplexity, while having a more stable training curve. We believe that this is because of the *softmax* in both the attention mechanism and the output layer of the transformer model. The performance of $PC_{\mathcal{F}_{KL}}$ is close to that of $BP$ and both training curves look stable. In practical terms, the language models trained by $BP$ and $PC_{\mathcal{F}_{KL}}$ do not differ significantly on the test set. In some cases, the predictions given by $PC_{\mathcal{F}_{KL}}$ are closer to the ground truth, e.g., for "Yet the bank and its executives are still ready to support specific Democratic [candidates]", $PC_{\mathcal{F}_{KL}}$ predicts "leaders" and "candidates" as top-2 choices. All models show failure in commonsense reasoning, e.g., for "I've been dreaming about this since I was a [child]" they fail to assign "child" with $> 1\%$ probability, which shows the limitations of small language models. More examples are given in the supplementary material.

## 5 Memory Consumption and Computational Complexity

**Memory consumption**: PC, in contrast to BP, stores $\mu$ and $\phi$ as different variables, which results in some memory overhead. On the contrary, the number of weights and training parameters used does not change between BP and PC. Therefore, if $M_{BP}$ is the memory consumption of training a model using BP, we have that, in general, $M_{PC} < 2 \cdot M_{BP}$. Actual values depend on the architecture and hyperparameters chosen.

**Computational complexity**: the complexity of a single forward pass in terms of the number of operations is comparable between PC and BP. The same can be said for the backward pass. However, in accordance with the EM algorithm, it is necessary to perform multiple updates on the neurons $x$ before updating the network weights $\theta$. This results in a multiplicative factor that can impact performance compared to BP. Nonetheless, from our experiments, we noticed that even a value as low as $T = 4$ or $T = 2$, where $T$ is the number of updates of the neurons before performing an update of the parameters, is sufficient given the right hyperparameters. In fact, the experiments on the transformer reached the best perplexity with exactly $T = 5$. Furthermore, we can take advantage of the features of PC. One of its major strengths is that each layer computation (both in the forward and backward pass) is local and, therefore, can be executed in parallel, removing one of the main bottlenecks of BP when training deep networks (i.e., the computations induced by each layer have to be executed sequentially). Thus, we expect PC to scale well on large architectures and to bring huge improvements on neuromorphic hardware. Finally, it has already been demonstrated that the speed of energy-based networks can be greatly increased by implementing the relaxation on analog hardware [Foroushani et al., 2020, Hertz et al., 1997], potentially resulting in energy-based networks being faster than BP. Thus, one scientific indication of this work is that the "analog-hardware-friendly" PC can have a reasonable performance on transformers, which opens the door to designing fast hardware-implemented transformers.

## 6 Related Work

In the last years, an active research direction that lies at the intersection of machine learning and cognitive science focuses on finding training algorithms for deep neural networks that have a degree of biological plausibility while obtaining good results on machine learning benchmarks. The most

popular ones are PC [Rao and Ballard, 1999, Whittington and Bogacz, 2017], equilibrium propagation [Scellier and Bengio, 2017, Scellier et al., 2018, Scellier and Bengio, 2019], and target propagation [Lee et al., 2015, Meulemans et al., 2020, Ernoult et al., 2022]. These methods share multiple similarities, both theoretically and in terms of performance. The first two methods, PC and equilibrium propagation, are able to approximate the weight update of BP when provided with a label that is close in distance to the neural activities of the last layer [Whittington and Bogacz, 2017, Scellier and Bengio, 2019]. Target propagation fails to have this property, but instead has been shown to approximate Gauss-Newton optimization [Meulemans et al., 2020]. However, PC possesses many unique properties that these methods lack. PC networks can in fact be used to produce efficient associative memory models [Salvatori et al., 2021], have an update mechanism that produces better learning properties than BP under specific conditions [Song et al., 2022], and allow training on graphs of any topology [Salvatori et al., 2022a]. Furthermore, they have achieved good results in classification [Han et al., 2018], generation [Ororbia and Kifer, 2020], and reinforcement learning Ororbia and Mali [2022a,b]. For a recent survey on these aspects, see [Millidge et al., 2022]. To conclude, we are not aware of any neuroscience-inspired learning method before this work that is able to generalize to complex tasks such as language modeling.

Progress in this direction is promising, as one of the main limitations of modern architectures is that they are extremely computationally expensive to be trained, with large-scale models sometimes requiring hundreds of GPUs for several weeks [Brown et al., 2020]. On the other hand, significant breakthroughs on neuromorphic and analog hardware have recently been achieved [Strukov et al., 2008, Sebastian et al., 2020], which can exploit the aforementioned properties of neuroscience-inspired learning methods, as shown in [Kendall et al., 2020], where the authors simulated the training of a multilayer network on an analog chip in an end-to-end fashion.

There has been a lot of research done towards bridging the gap in performance between state-of-the-art deep learning methods and neuroscience-inspired learning. Both fields can benefit from each other by drawing inspiration from each other's techniques. In neuroscience, understanding how the brain learns to associate different areas (e.g., visual and motor cortices) to successfully drive behaviour is of fundamental importance [Petreanu et al., 2012, Manita et al., 2015, Makino and Komiyama, 2015, Poort et al., 2015, Pakan et al., 2016, Zmarz and Keller, 2016, Attinger et al., 2017]. However, how to correctly modify synapses to achieve this has puzzled neuroscientists for decades. This is often referred to as the synaptic credit assignment problem [Rumelhart et al., 1986, Sutton and Barto, 1998, Roelfsema and van Ooyen, 2005, Bengio, 2014, Lee et al., 2015, Roelfsema and Holtmaat, 2018], for which the BP algorithm provides an elegant solution.

# 7 Conclusion

The main motivation behind this work was to make PC competitive with BP for complex deep neural architectures. The tasks in this work are among the most popular and important in the field: image generation and language modelling. In the first case, we trained a variational autoencoder. This model is fully Gaussian, but the bottleneck requires explicitly computable variances. While variations of PC with trainable variances are already defined in the literature [Millidge et al., 2021], they do not allow dependencies between the variance and the input. Rather, they act as a regulariser within the network. Consequently, they have not been used as a sampling scheme in a specific layer of a PC network. In the second case, we trained a transformer model, intractable before by PC networks, due to the presence of attention (and hence *softmax*), and showed results comparable to those of BP. Future work includes applying this method to other complex deep learning architectures, with the far-reaching goal of scaling PC to large-scale machine learning tasks and hence further closing the gap with BP-based learning.

# Acknowledgments

This work was supported by the Alan Turing Institute under the EPSRC grant EP/N510129/1, by the AXA Research Fund, the EPSRC grant EP/R013667/1, the MRC grant MC_UU_00003/1, the BBSRC grant BB/S006338/1, and by the EU TAILOR grant. We also acknowledge the use of the EPSRC-funded Tier 2 facility JADE (EP/P020275/1) and GPU computing support by Scan Computers International Ltd. Yuhang Song was supported by the China Scholarship Council under the State Scholarship Fund and by a J.P. Morgan AI Research Fellowship.

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
