## A $\mathcal{F}_{KL}$ Reduces to $\mathcal{F}$ under Gaussian Assumptions

In this section, we prove that $PC_{\mathcal{F}_{KL}}$ is a generalized version of PC and, therefore, it retains all its properties and previously achieved results when assuming a Gaussian generative model. Particularly, we show that the newly introduced energy function $\mathcal{F}_{KL}$ reduces to the PC original energy formulation $\mathcal{F}$ and, almost exactly, $\widetilde{\mathcal{F}}$, when reintroducing the Gaussian assumptions for the generative model. Consider the KL divergence formulation and a layer $l$. As per Eq. (9), $\mathcal{E}_l := D_{KL}[\mathcal{X}_l(\phi_l)||\widehat{\mathcal{X}}_l(\mu_l)]$. If we assume that $\mathcal{X}_l$ and $\widehat{\mathcal{X}}_l$ are multivariate Gaussian distributions with means $u_l$, $\widehat{u}_l$ and fixed diagonal covariance matrices $\Sigma_l$, $\widehat{\Sigma}_l$, we have that

$$\mathcal{E}_l = \frac{1}{2}(\sum_{i=1}^{w_l} \frac{\Sigma_{l,i} + (u_{l,i} - \widehat{u}_{l,i})^2}{\widehat{\Sigma}_{l,i}} + \ln \frac{\widehat{\Sigma}_{l,i}}{\Sigma_{l,i}} - 1). \tag{11}$$

By setting $\Sigma_l = \widehat{\Sigma}_l = I$, $u_l = \phi_l$, and $\widehat{u}_l = \mu_l$, then

$$2\mathcal{E}_l = \sum_{i=1}^{w_l} (u_{l,i} - \widehat{u}_{l,i})^2 = (\mu_l - \phi_l)^2 = \epsilon_l^2, \tag{12}$$

which equals the energy function for each layer used in Eq. (6). If, instead, we assume that $\widehat{\Sigma}_l$ is a learnable parameter associated with layer $l$ (that is, $\widehat{\Sigma}_l \in \theta_l$, while keeping $\Sigma_l = I$), we obtain:

$$\begin{aligned} 2\mathcal{E}_l &= (\sum_{i=1}^{w_l} \frac{\Sigma_{l,i} + (u_{l,i} - \widehat{u}_{l,i})^2}{\widehat{\Sigma}_{l,i}} + \ln \widehat{\Sigma}_{l,i} - 1) \\ &= (\sum_{i=1}^{w_l} \widehat{\Sigma}_{l,i}^{-1}(\epsilon_{l,i}^2 + 1) + \ln \widehat{\Sigma}_{l,i}) + k' = (\sum_{i=1}^{w_l} \widehat{\Sigma}_{l,i}^{-1}(\epsilon_{l,i}'^2) + \ln \widehat{\Sigma}_{l,i}) + k', \end{aligned} \tag{13}$$

where $k'$ is a constant. By substituting $(\epsilon_{l,i}^2 + 1) = \epsilon_{l,i}'^2$, $\mathcal{E}_l$ corresponds to the layer energy function obtained in Eq. (5).

## B Biological Plausibility

Biological plausibility is a generic concept in the literature, often used to state that a specific family of models behaves similarly to biological neural networks present in our brains. However, different definitions of biological plausibility exist in the literature, and a model can be considered biologically plausible according to some definitions and not others. In what follows, we refer to the definition introduced in [Whittington and Bogacz, 2017], mostly restricted to local computations and plasticity. We now discuss how our framework fails to satisfy them at the most general level, and how to address this limitation. Particularly, the learning dynamics determined by the $\mathcal{F}_{KL}$ energy respects the PC assumptions defined by Whittington and Bogacz [2017] at the layer level. Particularly:

- *Local computation*: the activity of each layer depends only on the activities of its input nodes and their synaptic weights (i.e., $\mu_l = f_l(\phi_{l-1}, \theta_l)$).

- *Local plasticity*: synaptic plasticity only depends on pre and post-synaptic nodes. In fact, to minimize $\mathcal{F}_{KL}$, we take the derivatives

$$\frac{\partial \mathcal{F}_{KL}}{\partial \theta} = \frac{\partial}{\partial \theta} \sum_{l=0}^{L} D_{KL}[\mathcal{X}_l(\phi_l^{\mathcal{D}})||\widehat{\mathcal{X}}_l(\mu_l^{\mathcal{D}})] = \sum_{l=0}^{L} \frac{\partial}{\partial \theta_l} D_{KL}[\mathcal{X}_l(\phi_l^{\mathcal{D}})||\widehat{\mathcal{X}}_l(\mu_l^{\mathcal{D}})],$$

and, analogously,

$$\begin{aligned} \frac{\partial \mathcal{F}_{KL}}{\partial \phi} = \sum_{l=0}^{L-1} \frac{\partial}{\partial \phi_l}(D_{KL}[\mathcal{X}_l(\phi_l^{\mathcal{D}})||\widehat{\mathcal{X}}_l(\mu_l^{\mathcal{D}})] + D_{KL}[X_{l+1}(\phi_{l+1}^{\mathcal{D}})||\widehat{\mathcal{X}}_{l+1}(\mu_{l+1}^{\mathcal{D}})]) \\ + \frac{\partial}{\partial \phi_L} D_{KL}[X_L(\phi_L^{\mathcal{D}})||\widehat{\mathcal{X}}_L(\mu_L^{\mathcal{D}})], \end{aligned}$$

where the terms of both summations only depends on $\phi_l$, $\phi_{l+1}$, and $\phi_{l-1}$.

However, this does not guarantee that the above two properties are satisfied at the neural level, as the exact neural circuit employed within each layer strictly depends on the distribution families chosen for $\widehat{\mathcal{X}}_l$ and $\mathcal{X}_l$. Consequently, while the original formulation of PC has some degree of biological plausibility, this may not be true in the general proposed framework. This is because we do not set any limit on the complexity of possible distributions. This could also have repercussions on eventual implementations on analog and neuromorphic hardware. Hence, an interesting open problem is understanding which classes of probability distributions are biologically plausible, and which allow our framework to be implemented on these emergent technologies. Researchers interested in developing biologically plausible models, could then only restrict their study to specific classes of probability distributions. The same applies to researchers interested in implementing models on analog circuits.

## C    A More Detailed Analysis of Learning Dynamics of $\mathcal{F}_{KL}$

In what follows, we explicitly derive the update rules for the two classes of distributions discussed in the main body of the paper: categorical distributions and Gaussian distributions with non-fixed variance.

**Categorical distributions**: A PC layer following a *softmax* activation function represents a categorical distribution over $w_l$ elements. Each node stores a different probability mass value. We have that, at each time step $t$:

$$
\begin{aligned}
\frac{\partial \phi_{l,j}}{\partial t} = -\frac{\partial \mathcal{F}_{KL}}{\partial \phi_{l,j}} &= -\frac{\partial}{\partial \phi_{l,j}}(\mathcal{E}_l + \mathcal{E}_{l+1}) \\
&= -\frac{\partial}{\partial \phi_{l,j}}\Big(\sum_{i=1}^{w_l}(\phi_{l,i}) \cdot \ln(\frac{\phi_{l,i}}{\mu_{l,i}}) + \mathcal{E}_{l+1}\Big) \\
&= -\ln\phi_{l,j} + \ln\mu_{l,j} - 1 - \frac{\partial \mathcal{E}_{l+1}}{\partial \phi_{l,j}},
\end{aligned}
\tag{14}
$$

where $\mathcal{E}_{l+1} = 0$ when $l = L$. Furthermore,

$$
\begin{aligned}
\frac{\partial \theta_{l,j,k}}{\partial t} = -\frac{\partial \mathcal{F}_{KL}}{\partial \theta_{l,j,k}} &= -\frac{\partial \mathcal{E}_l}{\partial \theta_{l,j,k}} \\
&= -\frac{\partial}{\partial \theta_{l,j,k}}\Big(\sum_{i=1}^{w_l}(\phi_{l,i}) \cdot \ln(\frac{\phi_{l,i}}{\mu_{l,i}})\Big) \\
&= \mu_{l,j}^{-1}\phi_{l,j}\,\frac{\partial \mu_{l,j}}{\partial \theta_{l,j,k}} \\
&= \begin{cases} (\phi_{l,j})(\mu_{l,j})(1 - \mu_{l,j}), & \text{if } j = k \\ -(\phi_{l,j})(\mu_{l,j})(\mu_{l,k}), & \text{otherwise} \end{cases},
\end{aligned}
\tag{15}
$$

where $\mu_l = f_l(\theta_l\,\phi_{l-1})$.

**Gaussian distributions**: As shown in Section 4.2, we can model a full Gaussian distribution $\mathcal{N}(\widehat{u}_l, \widehat{\Sigma}_l)$, with $(\widehat{u}_l, \widehat{\Sigma}_l) = \mu_l = f_l(\phi_{l-1}, \theta_l)$. In this scenario, the layer $l$ parameterises the distribution $\mathcal{N}(u_l, \Sigma_l)$, and $\phi_l = (u_l, \Sigma_l)$. We are, again, assuming diagonal covariance matrices. The dynamics are as follows:

$$
\begin{aligned}
\frac{\partial u_{l,j}}{\partial t} = -\frac{\partial \mathcal{F}_{KL}}{\partial u_{l,j}} &= -\frac{\partial}{\partial u_{l,j}}(\mathcal{E}_l + \mathcal{E}_{l+1}) \\
&= -\widehat{\Sigma}_{l,j}^{-1}\epsilon_{l,j} - \frac{\partial \mathcal{E}_{l+1}}{\partial u_{l,j}},
\end{aligned}
\tag{16}
$$

$$
\begin{aligned}
\frac{\partial \Sigma_{l,j}}{\partial t} = -\frac{\partial \mathcal{F}_{KL}}{\partial \Sigma_{l,j}} &= -\frac{\partial}{\partial \Sigma_{l,j}}(\mathcal{E}_l + \mathcal{E}_{l+1}) \\
&= \frac{1}{2}(\Sigma_{l,j}^{-1} - \widehat{\Sigma}_{l,j}^{-1}) - \frac{\partial \mathcal{E}_{l+1}}{\partial \Sigma_{l,j}},
\end{aligned}
\tag{17}
$$

and

$$\frac{\partial \theta_{l,j,k}}{\partial t} = -\frac{\partial \mathcal{F}_{KL}}{\partial \theta_{l,j,k}} = -\frac{\partial \mathcal{E}_l}{\partial \theta_{l,j,k}}$$

$$= \begin{cases} \widehat{\Sigma}_{l,j}^{-1} \epsilon_{l,j} - \frac{\partial \widehat{u}_{l,j}}{\partial \theta_{l,j,k}} & \text{if } j \leq w_l/2 \\ \frac{1}{2}\widehat{\Sigma}_{l,j'}^{-2}(\epsilon_{l,j'}^2 + \Sigma_{l,j'} - \widehat{\Sigma}_{l,j'}) - \frac{\partial \widehat{\Sigma}_{l,j'}}{\partial \theta_{l,j,k}} & \text{otherwise} \end{cases} \tag{18}$$

$$= \begin{cases} \widehat{\Sigma}_{l,j}^{-1} \epsilon_l - \frac{\partial f_{l,j}}{\partial \theta_{l,j,k}} & \text{if } j \leq w_l/2 \\ \frac{1}{2}\widehat{\Sigma}_{l,j'}^{-2}(\epsilon_{l,j'}^2 + \Sigma_{l,j'}) - \widehat{\Sigma}_{l,j'}^{-1} - \frac{\partial f_{l,j}}{\partial \theta_{l,j,k}} & \text{otherwise} \end{cases},$$

where $\epsilon_l = (u_l - \widehat{u}_l)$ and $j' = j - w/2$.

# D  Derivations of the Equations Used in this Work

In this section, we provide more explicit derivations for several of the equations presented in this work. By doing so, we hope to ease a detailed understanding of the mathematical framework that we defined.

- **Eq. (5):**

$$\widetilde{\mathcal{F}} = -\mathbb{E}_{q_\phi(x_{0:L}|d,o)}[\ln p(x_{0:L})] = \sum_{l=0}^{L} -\ln p(\phi_l|\mu_l) \quad \text{// Dirac-delta posterior and Eq. (4)}$$

$$= -\sum_{l=0}^{L} \ln \mathcal{N}(\phi_l; \mu_l, \Sigma_l) \qquad\qquad \text{// Gaussian generative model}$$

$$= \frac{1}{2}(\sum_{l=0}^{L} \epsilon_l^T \Sigma_l^{-1} \epsilon_l + \ln 2\pi|\Sigma_l|) \qquad\qquad \text{// } \epsilon_l = \phi_l - \mu_l$$

$$= \frac{1}{2}(\sum_{l=0}^{L} \sum_{i=1}^{w_l} \Sigma_{l,i}^{-1} \epsilon_{l,i}^2 + \ln \Sigma_{l,i}) + k. \qquad\qquad \text{// } \Sigma_l \text{ is a diagonal matrix}$$

- **Eq. (8):**

$$\bar{\mathcal{E}}_l = -\ln p(\phi_l^{\mathcal{P}}|\mu_l^{\mathcal{P}})$$

$$:= -\frac{1}{N}\sum_{i=1}^{N} \ln p(s_l^{(i)}|\mu_l^{\mathcal{P}}) \qquad\qquad \text{// by definition}$$

$$\approx -\mathbb{E}_{s_l \sim \mathcal{X}_l(\phi_l^{\mathcal{P}})}[\ln p(\widehat{\mathcal{X}}_l(\mu_l^{\mathcal{P}}) = s_l)] \qquad\qquad \text{// assuming large N}$$

$$= -\int_{s_l \in dom(\mathcal{X}_l(\phi_l^{\mathcal{P}}))} p(\mathcal{X}_l(\phi_l^{\mathcal{P}}) = s_l) \ln p(\widehat{\mathcal{X}}_l(\mu_l^{\mathcal{P}}) = s_l) \, ds_l$$

$$= \mathcal{H}(\mathcal{X}_l(\phi_l^{\mathcal{P}}), \widehat{\mathcal{X}}_l(\mu_l^{\mathcal{P}})). \qquad\qquad \text{// definition of } \mathcal{H}$$

- **Eq. (11)**

$$\mathcal{E}_l = D_{KL}[\mathcal{X}_l(\phi_l^{\mathcal{P}})||\widehat{\mathcal{X}}_l(\mu_l^{\mathcal{P}})]$$

$$= D_{KL}[\mathcal{N}(u_l, \Sigma_l)||\mathcal{N}(\widehat{u}_l, \widehat{\Sigma}_l)]$$

$$= -\int \mathcal{N}(x; u_l, \Sigma_l) \ln \mathcal{N}(x; \widehat{u}_l, \widehat{\Sigma}_l) \, dx + \int \mathcal{N}(x; u_l, \Sigma_l) \ln \mathcal{N}(x; u_l, \Sigma_l) \, dx$$

$$= \sum_{i=1}^{w_l} \frac{1}{2}\ln(2\pi\widehat{\Sigma}_l^2) + \frac{\Sigma^2 + (u - \widehat{u})^2}{2\widehat{\Sigma}_l^2} - \frac{1}{2}(1 + \ln(2\pi\Sigma_l^2)) \quad \text{// diagonal covariance matrices}$$

$$= \frac{1}{2}(\sum_{i=1}^{w_l} \frac{\Sigma_{l,i} + (u_{l,i} - \widehat{u}_{l,i})^2}{\widehat{\Sigma}_{l,i}} + \ln \frac{\widehat{\Sigma}_{l,i}}{\Sigma_{l,i}} - 1).$$

# E   Implementation Details

In this section, we provide a detailed description of the models and parameters needed to reproduce the results presented in this work. Note that our goal was to compare the performance of different training methods. Hence, we do not aim for state-of-the-art results, but rather a comparable performance across the different training methods for each employed architecture.

## E.1   Classification Networks

We used fully connected feedforward networks composed by a sequence of $L \in \{3, 4, 5\}$ fully connected layers of width $w \in \{256, 512, 1024\}$. The weights learning rate was set to $\beta_\theta = 0.0001$. We also experimented with different node learning rates $\beta_\phi \in \{0.01, 0.05, 0.025\}$. We used $T = 32$ $\phi$-steps and initialized the node values at $t = 0$ using a forward pass, as suggested by Song et al. [2020]. We used the Adam optimizer to optimize the weights of the model, while we used a stochastic gradient descent optimizer for the nodes $x$. We did not find any relevant differences in the observed relative performance of the three learning methods among the various combinations of hyperparameters tested. The results reported in Fig. 3 were obtained with $w = 512$, $L = 3$, and $\beta_\phi = 0.05$. We found that using *ReLU* instead of *tanh* as activation function significantly reduces the accuracy achieved by PC (at least with the highly-specific architectures used for this task).

## E.2   Variational Autoencoders

We used fully connected layers for both the encoders and the decoders. We trained several models with $L \in \{2, 3\}$ layers for both encoder and decoder and width $w \in \{256, 512\}$. We used 32 latent units for the bottleneck layer, divided equally to store mean and variance. The activation function used was *tanh*. Learning rates and optimizers are the same used for classification networks. The variance in the results reported is due to different combinations of the hyperparameters chosen to obtain one or the other architecture. In Fig. 5, we reported the learning curves for two models. The choice was completely random to highlight the comparable performance of BP and PC on a general architecture.

## E.3   Transformer Language Models

The 8001-token vocabulary is automatically generated based on a portion of the training data and includes the <sos>, <eos>, and <pad> tokens. The input of the model is restricted to sequences of length up to 34, where to the token of the sentence, we prepend the <sos> token and append the <eos> token. The tokens for each batch are further appended to the same length via the <pad> token.

To optimize the weights of the model, the AdamW optimizer is used with default (0.01) weight decay, and each model is trained for two epochs with a batch size of 8. We use a stochastic gradient descent optimizer for the nodes $x$.

Here are the hyperparameter ranges and best values used for each model:

For $BP$: $\beta_\theta \in \{0.0004, 0.0008, 0.0016, 0.0032, 0.0064\}$. Best value: 0.0016.

For $PC_\mathcal{F}$: $T \in \{4, 5, 6, 7, 8\}$, $\beta_\phi \in \{0.001953125, 0.00390625, 0.0078125, 0.015625, 0.03125\}$, $\beta_\theta \in \{0.0002, 0.0004, 0.0008, 0.0016, 0.0032, 0.0064, 0.0128\}$. Best values: $T = 4$, $\beta_\phi = 0.015625$, $\beta_\theta = 0.0064$.

For $PC_{\mathcal{F}_{KL}}$: $T \in \{4, 5, 6, 7, 8\}$, $\beta_\phi \in \{0.25, 0.5, 1.0\}$, $\beta_\theta \in \{0.000025, 0.00005, 0.0001, 0.0002, 0.0004, 0.0008, 0.0016\}$. Best values: $T = 5$, $\beta_\phi = 0.5$, $\beta_\theta = 0.0008$.

The total training time of the hyperparameter search is approximately 94 hours on one Nvidia Titan RTX GPU.

### E.3.1   Qualitative Results

Table 2 shows example sentence completions given by $BP$, $PC_\mathcal{F}$, and $PC_{\mathcal{F}_{KL}}$ along with the probabilities assigned to each prediction. The sentences were selected subjectively from the test dataset based on how interesting they are and cut right before a subjectively interesting word to be predicted.

| Input sentence | $BP$ | | $PC_{\mathcal{F}}$ | | $PC_{\mathcal{F}_{KL}}$ | |
|---|---|---|---|---|---|---|
| Yet the bank and its executives are still ready to support specific Democratic [candidates] | leaders | (7.5) | . | (1.0) | leaders | (12.1) |
| | Party | (7.2) | , | (1.0) | candidates | (7.3) |
| | candidates | (3.8) | and | (0.6) | presidential | (4.8) |
| GMAC started out offering car [loans] | and | (2.2) | , | (5.3) | sales | (4.4) |
| | sales | (1.7) | and | (3.1) | products | (2.3) |
| | , | (1.7) | in | (2.5) | services | (2.0) |
| I've been dreaming about this since I was a [child] | great | (1.6) | lot | (1.1) | " | (1.9) |
| | " | (1.5) | good | (1.1) | good | (1.2) |
| | good | (1.2) | very | (0.9) | year | (1.1) |
| Here is a breakdown of the seven taxes and fees that have been [collected] | a | (2.3) | a | (4.7) | a | (4.3) |
| | to | (2.0) | the | (2.3) | to | (2.6) |
| | in | (1.9) | in | (1.3) | the | (2.4) |
| Under the plan, Iceland will reimburse the [money] | first | (1.9) | best | (1.7) | first | (1.9) |
| | world | (1.0) | first | (1.6) | same | (1.2) |
| | same | (0.8) | most | (0.7) | world | (1.2) |
| Aniston and Pitt were still married when Pitt and Jolie made the 2005 [film] | , | (2.3) | , | (23.0) | . | (10.2) |
| | . | (1.9) | and | (5.6) | , | (10.0) |
| | World | (1.6) | . | (5.1) | and | (3.9) |

Table 2: Top predictions of each model for completing several sentences. The ground-truth completion is given in [brackets]; the model prediction format is: <word> (<probability %>).