# OpenReview forum: "Predictive Coding beyond Gaussian Distributions"
_NeurIPS.cc/2022/Conference — NeurIPS 2022 Accept_

### Official Review · Reviewer_3JGX · 2022-06-26

**Rating:** 7
**Confidence:** 5
**Soundness:** 4 excellent
**Presentation:** 4 excellent
**Contribution:** 4 excellent

**Summary:**

This work addresses the problem of generalizing Predictive Coding (PC) to arbitrary distributions. The related literature has presented PC as a biologically plausible alternative of backprop that relies on local (layerwise) updates. However, PC has so far been established as a credible alternative to backprop only when considering Gaussian activations. This is extremely unfortunate, as it limits the applicability of PC to virtually a small fraction of modern deep nets.

This paper generalizes PC to deal with any distribution, and importantly intractable distributions, approximated via a sampling scheme. The resulting framework is coherent with the PC theory, as it enables the definition of layer-wise energy functions that represent prediction errors.

**Questions:**

I need some deeper discussions on the weaknesses of the method compared to backprop. For instance, what happens in terms of computational complexity? This must be analyzed in terms of training time, inference time, and required resources (FLOPS). Do you foresee architectures where the proposed method may linger compared to BP? What are the directions for future research to address these limitations?

**Strengths And Weaknesses:**

Pros: Strong motivation; a very good mathematical foundation; sound mathematical proofs of on the equivalence with BP; potential for strong impact on the community; convincing simulations.
Cons: I need some deeper discussions on the weaknesses of the method compared to backprop. For instance, what happens in terms of computational complexity?

---

> ### Author Response · Authors · 2022-08-02
> **Response to Reviewer 3JGX (2/2)**
>
> - [computational complexity]: To conclude, there are reasons to bypass the current “efficiency” argument, as it is done in most of the papers that present neuroscience-inspired training methods for deep neural networks, such as equilibrium and target propagation [1,2,3], which are influential neuroscience-inspired methods to train deep neural networks, heavily pushed by people such as Yoshua Bengio. All these algorithms are much slower than backpropagation; however, they are extremely promising, as they would allow deep neural networks to be trained in an end-to-end fashion on physical chips, such as analog/photonic/quantum models [4,5,6]. Predictive coding is no different, with the big advantage that its important connections with neuroscientific theories, and promising performance obtained in multiple tasks (such as this one on language models), could make it the main candidate when it comes to training deep architectures on these physical chips. Progress in this direction would allow to (1) train large neural networks with little energy consumption: they would consume multiple orders of magnitude less energy than GPUs, (2) while being much faster to train. It is also expected that these chips, when (and if) developed, will be cheaper to build with respect to GPUs, hence (3) making computing power more accessible. There is still much research to be performed in this direction, especially from the hardware side, however, PC could play a large role the day we will manage to build these chips. In summary, as usual for papers on neuroscience-inspired training methods, the gap in efficiency on current hardware against backpropagation-based models should not be the deciding factor for the quality of a paper.
>
> [1] Scellier, Benjamin, and Yoshua Bengio. "Equilibrium propagation: Bridging the gap between energy-based models and backpropagation." Frontiers in computational neuroscience 11 (2017): 24.
>
> [2] Lee, Dong-Hyun, et al. "Difference target propagation." Joint European conference on machine learning and knowledge discovery in databases. Springer, Cham, 2015.
>
> [3] Bengio, Yoshua. "Deriving differential target propagation from iterating approximate inverses." arXiv preprint arXiv:2007.15139 (2020).
>
> [4] Zoppo, Gianluca, Francesco Marrone, and Fernando Corinto. "Equilibrium propagation for memristor-based recurrent neural networks." Frontiers in neuroscience 14 (2020): 240.
>
> [5] Kendall, Jack, et al. "Training end-to-end analog neural networks with equilibrium propagation." arXiv preprint arXiv:2006.01981 (2020).
>
> [6] Scellier, B., Mishra, S., Bengio, Y., & Ollivier, Y. (2022). Agnostic Physics-Driven Deep Learning. arXiv preprint arXiv:2205.15021.
>
> [7] Foroushani, A. N., Assaf, H., Noshahr, F. H., Savaria, Y. & Sawan, M. Analog circuits to accelerate the relaxation process in the equilibrium propagation algorithm. In 2020 IEEE International Symposium on Circuits and Systems (ISCAS), 1–5, IEEE, 2020.
>
> [8] Hertz, J., Krogh, A., Lautrup, B. & Lehmann, T. Nonlinear backpropagation: Doing backpropagation without derivatives of the activation function. IEEE Transactions on Neural Networks 8, 1321–1327, 1997.

---

> ### Author Response · Authors · 2022-08-02
> **Response to Reviewer 3JGX (1/2)**
>
> We thank the reviewer for the comments.
>
> ---
> > What are the weaknesses of the method compared to backprop?
>
> One of the main difficulties that we experienced is the ability to provide a formal analysis of the dynamics and convergence of the method without adding additional constraints. We have multiple interesting techniques to apply to a network, however, other than having plenty of empirical results, it’s hard to provide theoretical proofs of their effectiveness.
> Furthermore, as we will discuss later, the current hardware/libraries are not suited for PC, resulting in significant overhead in both programming and executing experiments. Our research is currently focusing on eliminating these barriers.
>
> ---
> > What are the computational costs?
>
> PC is an alternative to BP to train neural networks and both can be applied alternatively to the same architecture. As a result, they behave very similarly in terms of memory consumption, scaling to large datasets and to complex models. There are, however, several factors to be considered in terms of both memory consumption and computational complexity.
>
> - Memory consumption: PC stores $\mu$ and $\phi$ as different variables (while in BP they coincide, since the activation of one layer is fed directly to the next one), which results in some memory overhead. On the contrary, the number of weights and training parameters used doesn't change between BP and PC, as the underlying architecture is the same. In summary, if $M_{BP}$ is the memory consumption of training a model using BP, we have that, theoretically, $M_{PC}$ < $2 M_{BP}$. Actual values depend on the architecture and hyperparameters chosen.
> - Computational complexity: the complexity of a single forward pass in terms of the number of operations is comparable between PC and BP. The same can be said for the backward pass. However, in accordance with the EM algorithm, it is necessary to perform multiple updates on the neurons $x$ before updating the network weights $\theta$. This results in a multiplicative factor that can impact performance compared to BP. Nonetheless, from our experiments, we noticed that even a value as low as T=4 or T=2 (where T is the number of neurons’ updates before performing a weight update) is sufficient and can actually improve the results compared to higher T values when paired with the right hyperparameters. In fact, the experiment with the transformer reached its best perplexity with exactly T=4. Understanding how to tune and systematically decrease the value of T is outside the scope of this paper and an interesting topic for future research. Additionally, we can take advantage of the features of PC. One of its major strengths is that each layer computation (both in the forward and backward pass) is local and, therefore, can be executed in parallel, removing one of the main bottlenecks of BP: it does not matter how powerful one’s hardware is – if one scales an architecture to a large number of layers, one is forced to execute them sequentially, limiting the amount of hardware resources that can be used concurrently. Thus, we expect PC to scale well on large architectures and to bring huge improvements on neuromorphic hardware. The experiments reported in the paper, however, did not benefit from this parallelization, mainly due to a bug in Pytorch [Pytorch streams API don't execute concurrently, However same code in CUDA does. · Issue #48279 · pytorch/pytorch (github.com)](https://github.com/pytorch/pytorch/issues/48279), [CUDA streams not running in parallel? - PyTorch Forums](https://discuss.pytorch.org/t/cuda-streams-not-running-in-parallel/59998), [CUDA streams not being executed concurrently · Issue #19103 · pytorch/pytorch (github.com)](https://github.com/pytorch/pytorch/issues/19103)]. Furthermore, it has already been demonstrated that the speed of energy-based networks can be greatly increased by implementing the relaxation on analog hardware [7,8], potentially resulting in energy-based networks being faster than backpropagation. Thus, one scientific indication of this work is to such that the “analog-hardware-friendly” predictive coding can have reasonable performance on transformers, which opens the door of designing fast hardware-implemented transformers.

---

### Official Review · Reviewer_ADdV · 2022-07-09

**Rating:** 4
**Confidence:** 3
**Soundness:** 2 fair
**Presentation:** 2 fair
**Contribution:** 2 fair

**Summary:**

The authors extend the predictive coding formulation of training neural networks to the setting of non-Gaussian densities for the internal and output units (and non-Dirac approximate posteriors), instead using the more general KL divergence formulation. The authors apply this to supervised classification, VAE training, and training transformers. The results are typically better, in terms of performance and efficiency, than the Gaussian-formulation of predictive coding training and roughly comparable with backpropagation.

**Questions:**

Why ignore the approximate posterior entropy in Eq. 7? Shouldn’t Eq. 8 contain an expectation over all approximate posterior dimensions? As currently written, this is a mean field assumption over layers, which I would imagine wouldn’t scale particularly well to large model architectures.

What is the relative training time for each algorithm?

Does this more generic formulation retain the biological plausibility benefits of the original formulation? It seems that you now need to obtain more general gradients at each layer, as opposed to the relatively simple form (errors) of the original formulation.

Are the internal activations sampled? This would yield a more expressive model as compared with a Dirac approximate posterior.

How does this method differ from target propagation?

**Limitations:**

Limitations are perhaps somewhat addressed. One aspect that the authors have not fully discussed is the increased training time required for these algorithms. As the authors state, they use 32 iterations to infer the activations of the network per training iteration. Naively, I would expect the predictive coding-trained networks to be 32 times slower than their backprop counterparts. Likewise, it’s unclear whether it’s biologically plausible to assume that we can optimize the more generic free energy in Eq. 8. While the authors do not explicitly make claims of biological plausibility, I see this as one potential limitation of moving away from the original predictive coding formulation.

**Strengths And Weaknesses:**

**Strengths**

*Originality*:
The paper is somewhat original, in that it generalizes previous approaches to training neural networks with predictive coding, moving to more general distributions and more general functions. I see this as somewhat useful, as it opens up the space of models that can be trained with predictive coding. However, I question whether this formulation isn’t already encapsulated within hierarchical latent variable models (i.e., hierarchical VAEs) or related alternatives to backprop, like target propagation.

*Quality*:
The paper is of reasonable quality. The authors take a somewhat established area, predictive coding-based alternatives to backprop, and generalize it to non-Dirac approximate posteriors (and non-Gaussian priors). They attempt to illustrate the key differences of their method using diagrams and mathematical expressions. Finally, they perform comparisons with the original predictive coding formulation and backprop in several settings. While I can see weaknesses in each of these areas, the paper, overall, has many of the structural elements of what could become a high-quality paper.

*Clarity*:
For the most part, I was able to follow the writing and mathematical notation (although see below). I appreciate that the authors included diagrams to illustrate the key differences between backprop vs. predictive coding and Gaussian vs. non-Gaussian (or rather, Dirac vs. non-Dirac) predictive coding formulations. With relatively minor improvements, I feel that these aspects of the paper could become quite clear.

*Significance*:
To the best of my knowledge, alternatives to backpropagation have been primarily restricted to supervised learning with fully-connected or convolutional model architectures. This paper presents a generalization of the original predictive coding training formulation, which is demonstrated in unsupervised settings and with the transformer model architecture. The fact that this method is able to roughly match backprop in these settings is at least moderately significant.

**Weaknesses**

*Originality*:
I found the originality of the paper to be somewhat lacking. The main novelty of the paper is in moving beyond the original assumptions of Dirac approximate posteriors. However, in variational autoencoders (VAEs), which are also based around Gaussian latent variable models, these assumptions aren’t even entertained; in practice, almost everyone uses Gaussian priors and approximate posteriors, or even more general distributions like normalizing flows. Thus, the formulation isn’t all that novel, considering that it is standard within other areas of machine learning. Further, given the lack of clarity around the presentation (see below), I found it difficult to determine whether this more general formulation retains the ‘biological plausibility’ / locality benefits of the original predictive coding formulation.

*Quality*:
Much of the weakness of the quality of the work stems from the weaknesses outlined in other areas.

Here, I will reiterate that the presentation and empirical evaluation could be improved. In terms of presentation, I would have preferred to see clearer notation and diagrams, as well as the details of training (e.g., with an algorithm box). For the empirical evaluation, I would have liked to see experiments with larger models, e.g., even just convolutional, more datasets, and additional baselines. The only real baseline is the original predictive coding formulation. Given that the presented formulation is strictly more expressive, it’s not all that surprising that the performance is matched or improved.

*Clarity*:
I found the clarity of the paper to be one of its major weaknesses.

I appreciate the fact that the authors included diagrams in Figures 1 and 2 to attempt to illustrate the differences between 1) predictive coding and backprop and 2) predictive coding with and without Gaussian assumptions. However, I found these diagrams difficult to interpret, so they didn’t add much to my understanding of the key differences. It’s not always clear what each of the arrows and shapes are meant to represent computationally. I also didn’t find the inconsistent vs. consistent dichotomy in Figure 2 to be clear — if I understand correctly, this is just pointing out the fact that typical predictive coding formulations assume a deterministic inference estimate, whereas one could assume other forms of distributions. Inconsistent, to me, implies that there is some type of mistake in the original formulation.

I found the notation to be difficult to follow at times. For instance, the authors use $\phi$ to denote the approximate posterior estimate, initially a point estimate of the latent variables. Later on, they switch to a notation in which $\phi^\mathcal{S}$ denotes this single estimate ($\mathcal{S}$ for sample) and $\phi^\mathcal{D}$ denotes a distribution. While I feel that I understand the main points that they authors are trying to convey, I have to imagine that there are easier ways to present the formulation, particularly for readers that are unfamiliar with predictive coding or alternatives to backpropagation. Instead, I would have kept a consistent notation for the approximate posterior estimate, showing that the standard formulation of predictive coding makes a particular assumption (Dirac), but this is either a) not expressive enough or b) not suitable for categorical/discrete variables, necessitating other distributions.

The details of the formulation are quite unclear to me. For instance, the authors allude to using the EM algorithm for optimizing both the generative model parameters and inference estimate. In the (typical?) case of Dirac inference estimates, Gaussian priors, and fully-connected networks, this seems fairly reasonable, as the gradient consists of the error at each layer, perhaps including the derivative of the non-linearity. However, in the more general case of non-Dirac inference estimates, non-Gaussian priors, and more general network architectures, it’s unclear what is included in these gradients, let alone whether calculating these gradients is biologically plausible. Further, the authors make several assumptions in their formulation (Eqs. 7 & 8) that seem poorly motivated. For instance, why the ‘energy’ at each layer is assumed to be the cross-entropy between the approximate posterior and prior, rather than the (correct) KL divergence. I suppose this makes sense in the case of Gaussians with fixed variance, but it seems unfounded in more general settings. Likewise, the free energy expression in Eq. 8 is a sum across layers of the network, completely neglecting any dependencies between layers — this is a fairly strong assumption.

The empirical setup is also unclear. I would have preferred to see a more in-depth description of the network architectures, ideally with clear diagrams. Further, some of the training details were quite unclear, e.g., whether or not the internal Gaussian activations were sampled, the time required for the 32 E-step iterations used for inference, etc. These are very practically important, as they determine the computational efficiency and expressiveness of the network and training algorithm.

*Significance*:
While I admit that training alternative network architectures with predictive coding is mildly significant, I did not find the results to be particularly compelling. And while I understand that the point of the paper is not to obtain state-of-the-art performance on benchmarks, the empirical setups investigated by the authors are almost too simplistic as to be interesting, e.g., classification with 3-layer fully-connected networks, vanilla VAEs. Admittedly, I’m not sure whether transformers have been trained with backprop alternatives, so this could be somewhat significant. However, given my confusion around the specifics of the actual implementation (e.g., how are gradients obtained for inference?), it’s unclear, for instance, whether the model actually retains the biological plausibility of the original predictive coding formulation. And, ultimately, the authors are only able to roughly match the performance of backprop in these fairly standard settings. That is, the authors have not demonstrated that this technique opens up any new training settings. Finally, the authors repeatedly motivate their technique with the idea of being able to apply these algorithms on analog devices, however, it’s entirely unclear to me whether this algorithm is any more suited for these computing architectures than backprop.

---

> ### Author Response · Authors · 2022-08-02
> **Response to Reviewer ADdV (5/5)**
>
> ---
> > Is PC more efficient than BP on analog hardware?
>
> As already mentioned, it has been demonstrated that the speed of energy-based networks can be greatly increased by implementing the relaxation on analog hardware [1, 2], potentially resulting in energy-based networks being faster than backpropagation. More in detail, in [4], the authors implement exact backpropagation on physical chips. They are the first ones to do so. However, the process is quite slow, as there is the need of a digital control signal at every layer of the network. This is due to the sequential structure of deep models, where every operation of a layer has to (1) 'wait' for the information of all the previous (or, following during the backward pass) layers, and (2) be saved in memory via a Von-Neumann digital device. The situation would be completely different if using methods that would allow to train neural networks end-to-end, i.e., without any digital component, on the same chip: in this case, the learning process would be much faster, and would not need any external control to be performed. Examples of models with the potential of being trained this way are predictive coding and target propagation [3].
>
> [1] Foroushani, A. N., Assaf, H., Noshahr, F. H., Savaria, Y. & Sawan, M. Analog circuits to accelerate the relaxation process in the equilibrium propagation algorithm. In 2020 IEEE International Symposium on Circuits and Systems (ISCAS), 1–5, IEEE, 2020.
>
> [2] Hertz, J., Krogh, A., Lautrup, B. & Lehmann, T. Nonlinear backpropagation: Doing backpropagation without derivatives of the activation function. IEEE Transactions on Neural Networks 8, 1321–1327, 1997.
>
> [3] Kendall, J., Pantone, R., Manickavasagam, K., Bengio, Y., & Scellier, B. (2020). Training End-to-End Analog Neural Networks with Equilibrium Propagation. arXiv.
>
> [4] Wright, L.G., Onodera, T., Stein, M.M. et al. Deep physical neural networks trained with backpropagation. Nature 601, 549–555 (2022).

---

> > ### Comment · Reviewer_ADdV · 2022-08-09
> > **Response to Authors**
> >
> > Thanks for your detailed response to my review. There appears to be confusion on both sides here, and to reflect this, I've decreased the confidence of my score from 4 to 3. My confusion may stem, in part, from being more familiar with predictive coding for the purpose of representation learning rather than network training. In this setting, the priors are often Gaussian, but the approximate posterior may be a delta (i.e., a deterministic maximum-a-posteriori estimate) or a full distribution (typically Gaussian). As I understood the paper, the main contribution was in replacing Gaussian priors and MAP approximate posteriors of internal nodes with more general distribution families. From the author response, it appears that this is not the case, and rather, the authors are *only* expanding the class of prior distribution families. This is, therefore, *less* of a contribution than I had originally thought. The fact that this wasn’t clear from the paper underscores the need to improve the clarity of the notation, figures, and text.
> >
> > The authors have misunderstood my critique regarding target propagation and other distribution families. My point is that predictive coding, generally, is a formulation of variational inference, which can be used for **representation learning** (e.g., VAEs) or **network training** (this paper). One of the main differences here is interpreting what counts as a “level” in the latent hierarchy: is it multiple network layers or just one? In uses of predictive coding-related ideas for representation learning (hierarchical VAEs), more flexible distribution families are standard. Although these do not appear to be used in network training formulations of predictive coding, their usage does not constitute an entirely novel idea, as they are already common elsewhere. Regarding target propagation: yes, this uses a separate set of weights, but this can be seen as approximating the gradients calculated within predictive coding. The fact that target propagation uses Gaussian densities somewhat undercuts this paper's claims of originality.
> >
> > I still find the tasks and models to be too toy-ish for demonstrating this method. I think this is a valid critique, as computation now scales linearly with the number of inference iterations. And for deeper models, I would imagine that inference convergence would require additional iterations. While this concern is not specific to this particular method, this may point to serious flaws in the predictive coding formulation of network training.
> >
> > Overall, I do not think this paper is ready for publication due to issues with originality, clarity, and significance. With all due respect to the other reviewers, I found the other reviews to be lacking in critical examination of the proposed method.

---

> > > ### Author Response · Authors · 2022-08-09
> > > **Response #2 to Reviewer ADdV**
> > >
> > > We thank the reviewer for the response.
> > >
> > > Response to:
> > >
> > >  paragraph 1: In most of deep learning (if not all given the nature of neural networks and backpropagation), the value contained in every single neuron is interpreted as the single point mass of the posterior distribution $p(x|d)$ (where $x$ represents the neurons and $d$ the input data). The way these values are interpreted by the model (i.e. as Gaussian distributions inside a VAE) is independent of the posterior chosen for each neuron and dependent, instead, on the architecture used. Therefore, we believe that our definition, and predictive coding in general (as we are not the one introducing this aspect), behaves analogously to other techniques.
> > >
> > >  paragraph 2: To reply to the reviewer's question: when we talk about the latent hierarchy in models we refer to the multiple layers. Regarding the remaining of the paragraph: PC is a training algorithm (at least according to the way we analyse it in this work) and, as such, it is compared to BP and other training techniques. The architecture and interpretation imposed on the neurons (as mentioned previously you can interpret the values in the bottleneck layer of a VAE as $\mu$ and $sigma$) is independent of the posterior chosen inside the training algorithm. We are not sure we are correctly understanding the raised concern. Regarding target propagation, it is unclear how the fact that "target propagation uses Gaussian densities somewhat undercuts this paper's claims of originality", as our work aims precisely at expanding PC to non-Gaussian distribution.
> > >
> > >  paragraph 3: Actually, our experiments are not at all "toy-ish": If you have a look at very recent published work on neuroscience-inspired alternatives to backpropagation, this is immediate to note.  In these works, the experiments are often limited to classification/generation tasks on very simple datasets, such as MNIST/FashionMNIST and, sometimes, CIFAR10. As a reference to works published in 2022 in influential journals/conference (ICML and Nature Comm.) by influential people (Bengio), see, for example, [1] or [2]).
> > >  On the contrary, we believe we can easily claim that we are the first ones **not** performing toy experiments, as we are the first ones managing to train language models with transformers via a neuroscience-inspired framework.
> > >
> > > [1] Maxence M Ernoult, Fabrice Normandin, Abhinav Moudgil, Sean Spinney, Eugene Belilovsky, Irina Rish, Blake Richards, Yoshua Bengio. Towards Scaling Difference Target Propagation by Learning Backprop Targets. Proceedings of the 39th International Conference on Machine Learning, PMLR 162:5968-5987, 2022.
> > >
> > > [2] Ororbia, A., Kifer, D. The neural coding framework for learning generative models. Nat Commun 13, 2064 (2022).

---

> ### Author Response · Authors · 2022-08-02
> **Response to Reviewer ADdV (4/5)**
>
> - [computational complexity]: To conclude, there are reasons to bypass the current “efficiency” argument, as it is done in most of the papers that present neuroscience-inspired training methods for deep neural networks, such as equilibrium and target propagation [1,2,3], which are influential neuroscience-inspired methods to train deep neural networks, heavily pushed by people such as Yoshua Bengio. All these algorithms are much slower than backpropagation; however, they are extremely promising, as they would allow deep neural networks to be trained in an end-to-end fashion on physical chips, such as analog/photonic/quantum models [4,5,6]. Predictive coding is no different, with the big advantage that its important connections with neuroscientific theories, and promising performance obtained in multiple tasks (such as this one on language models), could make it the main candidate when it comes to training deep architectures on these physical chips. Progress in this direction would allow to (1) train large neural networks with little energy consumption: they would consume multiple orders of magnitude less energy than GPUs, (2) while being much faster to train. It is also expected that these chips, when (and if) developed, will be cheaper to build with respect to GPUs, hence (3) making computing power more accessible. There is still much research to be performed in this direction, especially from the hardware side, however, PC could play a large role the day we will manage to build these chips. In summary, as usual for papers on neuroscience-inspired training methods, the gap in efficiency on current hardware against backpropagation-based models should not be the deciding factor for the quality of a paper.
>
> [1] Scellier, Benjamin, and Yoshua Bengio. "Equilibrium propagation: Bridging the gap between energy-based models and backpropagation." Frontiers in computational neuroscience 11 (2017): 24.
>
> [2] Lee, Dong-Hyun, et al. "Difference target propagation." Joint European conference on machine learning and knowledge discovery in databases. Springer, Cham, 2015.
>
> [3] Bengio, Yoshua. "Deriving differential target propagation from iterating approximate inverses." arXiv preprint arXiv:2007.15139 (2020).
>
> [4] Zoppo, Gianluca, Francesco Marrone, and Fernando Corinto. "Equilibrium propagation for memristor-based recurrent neural networks." Frontiers in neuroscience 14 (2020): 240.
>
> [5] Kendall, Jack, et al. "Training end-to-end analog neural networks with equilibrium propagation." arXiv preprint arXiv:2006.01981 (2020).
>
> [6] Scellier, B., Mishra, S., Bengio, Y., & Ollivier, Y. (2022). Agnostic Physics-Driven Deep Learning. arXiv preprint arXiv:2205.15021.
>
> [7] Foroushani, A. N., Assaf, H., Noshahr, F. H., Savaria, Y. & Sawan, M. Analog circuits to accelerate the relaxation process in the equilibrium propagation algorithm. In 2020 IEEE International Symposium on Circuits and Systems (ISCAS), 1–5, IEEE, 2020.
>
> [8] Hertz, J., Krogh, A., Lautrup, B. & Lehmann, T. Nonlinear backpropagation: Doing backpropagation without derivatives of the activation function. IEEE Transactions on Neural Networks 8, 1321–1327, 1997.
>
> ---
> > How does the method compare to Target propagation?
>
> Note that this answer is independent from our work, as our generalization of PC does not make it more/less similar to it. Target propagation, as the name suggests, computes a target on the output layer that is then “propagated back” by a set of weights that are different from the feedforward ones. Here are two differences: (1) it has two weight matrices per layer: a feedforward and a backward one; and (2) the target of every layer (that can be seen as similar to the value nodes of PC) is propagated via a backward pass, and not computed via an energy minimization process. The similarity stands in the fact that the weight update is then performed to minimize a local error, given by the difference between the target and the actual value of a layer.

---

> ### Author Response · Authors · 2022-08-02
> **Response to Reviewer ADdV (3/5)**
>
> ---
> > What are the computational costs?
>
> One of the main difficulties that we experienced is the ability to provide a formal analysis of the dynamics and convergence of the method without adding additional constraints. We have multiple interesting techniques to apply to a network, however, other than having plenty of empirical results, it’s hard to provide theoretical proofs of their effectiveness.
> Furthermore, as we will discuss later, the current hardware/libraries are not suited for PC, resulting in significant overhead in both programming and executing experiments. Our research is currently focusing on eliminating these barriers.
>
> PC is an alternative to BP to train neural networks and both can be applied alternatively to the same architecture. As a result, they behave very similarly in terms of memory consumption, scaling to large datasets and to complex models. There are, however, several factors to be considered in terms of both memory consumption and computational complexity.
>
> - Memory consumption: PC stores $\mu$ and $\phi$ as different variables (while in BP they coincide, since the activation of one layer is fed directly to the next one), which results in some memory overhead. On the contrary, the number of weights and training parameters used doesn't change between BP and PC, as the underlying architecture is the same. In summary, if $M_{BP}$ is the memory consumption of training a model using BP, we have that, theoretically, $M_{PC}$ < $2 M_{BP}$. Actual values depend on the architecture and hyperparameters chosen.
> - Computational complexity: the complexity of a single forward pass in terms of the number of operations is comparable between PC and BP. The same can be said for the backward pass. However, in accordance with the EM algorithm, it is necessary to perform multiple updates on the neurons $x$ before updating the network weights $\theta$. This results in a multiplicative factor that can impact performance compared to BP. Nonetheless, from our experiments, we noticed that even a value as low as T=4 or T=2 (where T is the number of neurons’ updates before performing a weight update) is sufficient and can actually improve the results compared to higher T values when paired with the right hyperparameters. In fact, the experiment with the transformer reached its best perplexity with exactly T=4. Understanding how to tune and systematically decrease the value of T is outside the scope of this paper and an interesting topic for future research. Additionally, we can take advantage of the features of PC. One of its major strengths is that each layer computation (both in the forward and backward pass) is local and, therefore, can be executed in parallel, removing one of the main bottlenecks of BP: it does not matter how powerful one’s hardware is – if one scales an architecture to a large number of layers, one is forced to execute them sequentially, limiting the amount of hardware resources that can be used concurrently. Thus, we expect PC to scale well on large architectures and to bring huge improvements on neuromorphic hardware. The experiments reported in the paper, however, did not benefit from this parallelization, mainly due to a bug in Pytorch [Pytorch streams API don't execute concurrently, However same code in CUDA does. · Issue #48279 · pytorch/pytorch (github.com)](https://github.com/pytorch/pytorch/issues/48279), [CUDA streams not running in parallel? - PyTorch Forums](https://discuss.pytorch.org/t/cuda-streams-not-running-in-parallel/59998), [CUDA streams not being executed concurrently · Issue #19103 · pytorch/pytorch (github.com)](https://github.com/pytorch/pytorch/issues/19103)]. Furthermore, it has already been demonstrated that the speed of energy-based networks can be greatly increased by implementing the relaxation on analog hardware [7,8], potentially resulting in energy-based networks being faster than backpropagation. Thus, one scientific indication of this work is to such that the “analog-hardware-friendly” predictive coding can have reasonable performance on transformers, which opens the door of designing fast hardware-implemented transformers.

---

> ### Author Response · Authors · 2022-08-02
> **Response to Reviewer ADdV (2/5)**
>
> ---
> > What are the details on the training algorithm?
>
> As we didn’t change the general structure of PC, but rather, expanded it, we are still using the default algorithm used in all the existing literature on the topic. We will add the details of it to the supplementary material.
>
> ---
> > Could you provide more empirical results?
>
> We showed with 3 very different tasks (supervised classification, unsupervised generation, and language modelling tasks) that our method is strictly (and significantly) outperforming the original one. This should be more than sufficient to show the effectiveness of our method, considering, in particular, that our goal was highlighting the scenarios in which the original PC fails to perform comparably to BP.
>
> ---
> >  Could you please provide clarifications on the notation and explanations?
>
> We agree that our notation lacks clarity. We addressed the problem by replacing the pictures with more explicit ones, removing unnecessary elements, and introducing a legend. We added a table to summarise the notation in appendix F; it will be integrated into the paper body. See the updated Figure 1. We will update figure 2 to match the new style
>
> ---
> > What do you mean when you refer to the inconsistency of the original PC formulation?
>
> The “problem” with the original formulation of PC is that the value $\phi$ of a PC layer is both interpreted (by different components) as distribution parameters or as a point sample. This is fine as long as we assume a Gaussian prior, since the two are interchangeable, however, this is not true for the majority of probability distributions. By pointing it out, we suggest an alternative formulation that, through an additional sampling phase, allows each layer to be interpreted exclusively as distribution parameters and, therefore, expanded to any possible distribution family.
>
> ---
> > What is the role of $\phi$ in the PC formulation?
>
> $\phi$ always represents the posterior approximation of our generative model. However, the way the value is interpreted within the network changes and can be either a sample or a distribution. In general, we employed a notation similar to the one used by other works in the literature to preserve consistency
>
> ---
> > Could you please clarify the details of the new formulation?
>
> As previously mentioned, we did not change the general mathematical formulation of the generative model and its posterior approximation. We only removed one of its assumptions and showed how to expand the energy function to reflect the change. As a consequence, all the details of the implementation/properties of PC are the same as previously (we already addressed the effects of a more general energy function in a previous point). Since you mentioned “non-Dirac inference estimates”, it is possible that we misunderstood the question, so please feel free to clarify it.
>
> ---
> > Why do you remove the posterior entropy in Equation 7 [resulting in KL divergence instead of cross entropy]?
>
> We opted for the KL divergence, since (as shown in Appendix A) it can be reduced to the original PC formulation under the same assumptions. We experimentally tried with both KL and cross-entropy and did not experience any difference. However, we cannot guarantee that one is better than the other, and we will keep studying and experimenting with both.
>
> ---
> > Is equation 8 still respecting all the requirements of PC for biological plausibility? Shouldn't there be an expectation other the posterior distribution in the equation?
>
> The equation represents the sum of the local energy of each layer $l$, which is the same formula used by the original formulation. We simply redefined the value of the probability $\ln p(\phi_l^D| \mu_l^D)$. Since both $\phi_l$ and $\mu_l$ are local values (they are indeed the same ones used in the original PC), the overall computation is analogous to the original PC, using only local gradients. Since we are using a Dirac posterior, the expectation collapses to a single value.
>
> ---
> > How are the gradients computed?
>
> The code was developed using Pytorch, therefore we used the standard autograd implementation to compute the gradients of the energy at each layer. However, due to the simplicity of the formulas involved, they can also be computed manually and included in the network.
>
> ---
> > What new training settings are possible with PC, compared than BP?
>
> Our work is using a more general framework with hopes to develop algorithms that are more expressive and efficient. For example, PC allows for loops within the layers of the models. There are also multiple ways in which a PC network can be trained and queried, among the many, for example, it is possible to perform both forward (e.g., predict digit given image) and backward (e.g., predict image given digit) inference within the same architecture [1].
>
> [1] Salvatori T., Pinchetti L., Millidge B., Song Y., Bogacz R., and Lukasiewicz T.. Learning on arbitrary graph topologies via predictive coding. arXiv:2201.13180, 2022a

---

> ### Author Response · Authors · 2022-08-02
> **Response to Reviewer ADdV (1/5)**
>
> We thank the reviewer for the comments.
>
> **Important points / Clearing misunderstandings:**
> - > *“The authors extend the predictive coding formulation of training neural networks to the setting of non-Gaussian densities for the internal and output units (and non-Dirac approximate posteriors)”*, *“The authors take a somewhat established area, predictive coding-based alternatives to backprop, and generalize it to non-Dirac approximate posteriors”*, …
>
>   Note that we did not change the approximate posterior family, since a Dirac distribution allows us to maintain a simple formulation for the energy function. Our work expanded the original formulation of PC beyond the assumption of using exclusively a Gaussian form for the generative model.
>
> - > *“However, I question whether this formulation isn’t already encapsulated within hierarchical latent variable models (i.e., hierarchical VAEs) or related alternatives to backprop, like target propagation”*
>
>   It is not: backpropagation, predictive coding, and target propagation are three distinct algorithms. Our proposed generalization version of PC does not change this, as our work focuses on PC, and does not bring it any further/closer to the other two algorithms. The fact that there exist architectures (such as VAEs) that rely on different distribution families is orthogonal to the training algorithm used. Indeed, in our work, we show that the original formulation of PC fails exactly when training those architectures.
>
> ---
> > Originality: The main novelty of the paper is in moving beyond the original assumptions of Dirac approximate posteriors.
>
> This is incorrect: we still use the original Dirac-delta formulation commonly used in the predictive coding framework (see, for example, [2] or analogous formulations such as [1]).
> In our work, we only challenge the Gaussian assumption for the generative model.
>
> [1] Buckley C. L., Kim C. S., McGregor S., and Seth A. K.. The free energy principle for action and perception: A mathematical review. Journal of Mathematical Psychology, 81:55–79, 2017.
>
> [2] Millidge B., Seth A. K., and Buckley C. L.. Predictive coding: a theoretical and experimental review. CoRR, abs/2107.12979, 2021.
>
> > Originality: almost everyone uses Gaussian priors and approximate posteriors, or even more general distributions.
>
> Note that this is exactly the strength of our work. As you correctly state, in standard deep learning it is common practice to use probability distributions that are more general than Gaussians. However, before our work, this was not possible using predictive coding networks: Our work is about showing the effectiveness compared to BP of a new and different training algorithm, not a different architecture. NFs and VAEs are architectures that have been trained exclusively with the BP training algorithm. As we showed in our work, the original formulation of PC is insufficient and not suited to train those models, since we prove that the distributions assumed at a structure level (e.g., which architecture we use to solve which problem), determine the energy function that we should use. This is a link that wasn’t previously clear.
>
> As pointed out by other reviewers, our approach finds empirical success exactly in those architectures in which we cannot assume Gaussian distributions for the underlying model.  “[Our work] points out a concrete scenario for which traditional PC fails (i.e. the softmax activation) and clearly demonstrates the improvement of the newly proposed method”.
>
> ---
> > Does the model retain the biological plausibility of the original PC formulation?
>
> As shown in Appendix B, the new proposed energy function is still local at a layer level, thus retaining the most appealing quality of PC. This is supported by other reviewers (e.g., “The resulting framework is coherent with the PC theory, as it enables the definition of layer-wise energy functions that represent prediction errors”).
> In particular, our formulation is equivalent to that of the original definition of PC when reintroducing a Gaussian form for the generative model, therefore, under those settings, it retains the biological plausibility of the original PC formulation. This result, however, does not generalize to every probability distribution: biological plausibility depends on the specific distribution used. It is hence not possible to answer this question for the most general case. However, in our work, we use networks with Gaussian or categorical (softmax) distributions. As it has been recently shown that there exists a biological implementation of the softmax function [1], models in our work do retain the biological plausibility of the original PC formulation.
>
> [1] Snow, Mallory A., and Jeff Orchard. "Biological Softmax: Demonstrated in Modern Hopfield Networks." Proceedings of the Annual Meeting of the Cognitive Science Society. Vol. 44. No. 44. 2022.

---

### Official Review · Reviewer_9uRC · 2022-07-11

**Rating:** 7
**Confidence:** 3
**Soundness:** 3 good
**Presentation:** 3 good
**Contribution:** 3 good

**Summary:**

This paper proposes a modification of the predictive coding (PC) paradigm in the context of variational inference with artificial neural networks. In particular, while the canonical PC framework assumes Gaussian form of the underlying generative model, the proposed PC modification extends PC to arbitrary distributions. The authors provide three sets of experiments which compare the proposed PC modification to two other methods of training, namely, backpropagation and regular PC. The empirical results suggest that the proposed PC modification outperforms regular PC on the task of classification for MNIST and CIFAR-10 and the task of language modeling. It is comparable to backpropagation in the classification of MNSIT and CIFAR-10 datasets and in training a VAE on MNIST data (in terms of reconstruction loss on the test set).


**Questions:**

1. Have the authors explored how the proposed method scales to larger architectures and datasets?

2. What is the relative memory and computational costs of PC vs BP in the three experimental setups?

3. In the experiments with VAEs, what are the two architectures used (line 241)?


**Limitations:**

The authors address limitations of their work. They mention that the KL divergence between $\chi_l$ and $\hat{\chi_l}$ should be computationally efficient to derive (line 171).
The authors also discuss the limitation of the biological plausibility of the proposed generalized PC in the Appendix - I believe this could be mentioned in the main text.
The authors could mention potential negative societal impacts, for example, in the case when the model might propagate harmful bias that is present in the training data (which is inherent to any machine learning system).


**Strengths And Weaknesses:**

Strengths

- The proposed method is well-motivated (addresses a limitation of the existing PC framework).
- The claims in the paper are supported with empirical evidence.
- The paper is well-organized and clearly written.

Room for improvement

- The paper could discuss the computational complexity and scalability of the proposed model.
- The authors could run multiple seeds in their experiments to provide error bars.

---

> ### Author Response · Authors · 2022-08-02
> **Response to Reviewer 9uRC (2/2)**
>
> - [computational complexity]: To conclude, there are reasons to bypass the current “efficiency” argument, as it is done in most of the papers that present neuroscience-inspired training methods for deep neural networks, such as equilibrium and target propagation [1,2,3], which are influential neuroscience-inspired methods to train deep neural networks, heavily pushed by people such as Yoshua Bengio. All these algorithms are much slower than backpropagation; however, they are extremely promising, as they would allow deep neural networks to be trained in an end-to-end fashion on physical chips, such as analog/photonic/quantum models [4,5,6]. Predictive coding is no different, with the big advantage that its important connections with neuroscientific theories, and promising performance obtained in multiple tasks (such as this one on language models), could make it the main candidate when it comes to training deep architectures on these physical chips. Progress in this direction would allow to (1) train large neural networks with little energy consumption: they would consume multiple orders of magnitude less energy than GPUs, (2) while being much faster to train. It is also expected that these chips, when (and if) developed, will be cheaper to build with respect to GPUs, hence (3) making computing power more accessible. There is still much research to be performed in this direction, especially from the hardware side, however, PC could play a large role the day we will manage to build these chips. In summary, as usual for papers on neuroscience-inspired training methods, the gap in efficiency on current hardware against backpropagation-based models should not be the deciding factor for the quality of a paper.
>
> ---
> > What is the architecture used for the VAEs?
>
> The details are explained in Appendix E.2, we have added more information: We used fully connected layers for both the encoders and the decoders. We trained several models
> with $L \in $ { $2, 3$ } layers for both encoder and decoder and width $w \in $ { $256, 512$ }. We used 32 latent units for the bottleneck layer, divided equally to store mean and variance. The activation function used was tanh. Learning rates and optimizers are the same used for the classification networks. The variance in the results reported is due to different combinations of the hyperparameters chosen to obtain one or the other architecture. The choice of the hyperparameters defining the architecture among the possible ones was completely random to highlight the comparable performance of BP and PC on a general architecture.
>
> ---
> > Confidence intervals are missing in the experimental section. Is it possible to add them?
>
> We have added a confidence interval to the transformer’s test results, as it is the most impactful result. In general, the added interval does not change the conclusion reached in the paper.
>
> |    Method    | Test perplexity ($\mu \pm \sigma$) |
> |----------------|------------------------------|
> | BP              | 162.64 $\pm$ 0.76     |
> | $PC_{F_{KL}}$ | 175.9081 $\pm$ 1.74 |
> | $PC$          | 590.08 $\pm$ 12.6     |
>
> ---
> > Are there potential negative societal impacts?
>
> As you suggested, the model has the same limitations as any machine learning system in relation to the bias present in the learning data. However, due to the more modular nature of PC (i.e., each layer is an independent unit), we hope it will be possible to have tools to better study and eventually eliminate the problem.
>
> [1] Scellier, Benjamin, and Yoshua Bengio. "Equilibrium propagation: Bridging the gap between energy-based models and backpropagation." Frontiers in computational neuroscience 11 (2017): 24.
>
> [2] Lee, Dong-Hyun, et al. "Difference target propagation." Joint European conference on machine learning and knowledge discovery in databases. Springer, Cham, 2015.
>
> [3] Bengio, Yoshua. "Deriving differential target propagation from iterating approximate inverses." arXiv preprint arXiv:2007.15139 (2020).
>
> [4] Zoppo, Gianluca, Francesco Marrone, and Fernando Corinto. "Equilibrium propagation for memristor-based recurrent neural networks." Frontiers in neuroscience 14 (2020): 240.
>
> [5] Kendall, Jack, et al. "Training end-to-end analog neural networks with equilibrium propagation." arXiv preprint arXiv:2006.01981 (2020).
>
> [6] Scellier, B., Mishra, S., Bengio, Y., & Ollivier, Y. (2022). Agnostic Physics-Driven Deep Learning. arXiv preprint arXiv:2205.15021.
>
> [7] Foroushani, A. N., Assaf, H., Noshahr, F. H., Savaria, Y. & Sawan, M. Analog circuits to accelerate the relaxation process in the equilibrium propagation algorithm. In 2020 IEEE International Symposium on Circuits and Systems (ISCAS), 1–5, IEEE, 2020.
>
> [8] Hertz, J., Krogh, A., Lautrup, B. & Lehmann, T. Nonlinear backpropagation: Doing backpropagation without derivatives of the activation function. IEEE Transactions on Neural Networks 8, 1321–1327, 1997.

---

> > ### Comment · Reviewer_9uRC · 2022-08-08
> > **Thank you for your response!**
> >
> > Thank you for the detailed and informative response! Also thank you for addressing my question about the VAEs architecture, adding confidence intervals, and commenting on the negative impacts. If the page limit allows it, I’d suggest including some more of the discussion on the memory consumption and computational complexity in the main text as I found it insightful and think it might be useful for others, too.

---

> > > ### Author Response · Authors · 2022-08-09
> > > **Thank you**
> > >
> > > We thank the reviewer for the time spend reading our rebuttals. We will find the space to add a discussion about the efficiency in the main body of the paper. We hope that the reviewer's concerns were addressed during the rebuttal, and if so, we kindly ask the reviewer to increase the score.

---

> > > > ### Comment · Reviewer_9uRC · 2022-08-09
> > > > **Concerns addressed - increasing score**
> > > >
> > > > I thank the authors for addressing my comments and questions! I have increased my score.

---

> ### Author Response · Authors · 2022-08-02
> **Response to Reviewer 9uRC (1/2)**
>
> We thank the reviewer for the comments.
>
> ---
> > How does the method scale to larger architectures and datasets, and what are the computational costs?
>
> PC is an alternative to BP to train neural networks and both can be applied alternatively to the same architecture. As a result, they behave very similarly in terms of memory consumption, scaling to large datasets and to complex models. There are, however, several factors to be considered in terms of both memory consumption and computational complexity.
>
> - Memory consumption: PC stores $\mu$ and $\phi$ as different variables (while in BP they coincide, since the activation of one layer is fed directly to the next one), which results in some memory overhead. On the contrary, the number of weights and training parameters used doesn't change between BP and PC, as the underlying architecture is the same. In summary, if $M_{BP}$ is the memory consumption of training a model using BP, we have that, theoretically, $M_{PC}$ < $2 M_{BP}$. Actual values depend on the architecture and hyperparameters chosen.
> - Computational complexity: the complexity of a single forward pass in terms of the number of operations is comparable between PC and BP. The same can be said for the backward pass. However, in accordance with the EM algorithm, it is necessary to perform multiple updates on the neurons $x$ before updating the network weights $\theta$. This results in a multiplicative factor that can impact performance compared to BP. Nonetheless, from our experiments, we noticed that even a value as low as T=4 or T=2 (where T is the number of neurons’ updates before performing a weight update) is sufficient and can actually improve the results compared to higher T values when paired with the right hyperparameters. In fact, the experiment with the transformer reached its best perplexity with exactly T=4. Understanding how to tune and systematically decrease the value of T is outside the scope of this paper and an interesting topic for future research. Additionally, we can take advantage of the features of PC. One of its major strengths is that each layer computation (both in the forward and backward pass) is local and, therefore, can be executed in parallel, removing one of the main bottlenecks of BP: it does not matter how powerful one’s hardware is – if one scales an architecture to a large number of layers, one is forced to execute them sequentially, limiting the amount of hardware resources that can be used concurrently. Thus, we expect PC to scale well on large architectures and to bring huge improvements on neuromorphic hardware. The experiments reported in the paper, however, did not benefit from this parallelization, mainly due to a bug in Pytorch [Pytorch streams API don't execute concurrently, However same code in CUDA does. · Issue #48279 · pytorch/pytorch (github.com)](https://github.com/pytorch/pytorch/issues/48279), [CUDA streams not running in parallel? - PyTorch Forums](https://discuss.pytorch.org/t/cuda-streams-not-running-in-parallel/59998), [CUDA streams not being executed concurrently · Issue #19103 · pytorch/pytorch (github.com)](https://github.com/pytorch/pytorch/issues/19103)]. Furthermore, it has already been demonstrated that the speed of energy-based networks can be greatly increased by implementing the relaxation on analog hardware [7,8], potentially resulting in energy-based networks being faster than backpropagation. Thus, one scientific indication of this work is to such that the “analog-hardware-friendly” predictive coding can have reasonable performance on transformers, which opens the door of designing fast hardware-implemented transformers.

---

### Official Review · Reviewer_Bynh · 2022-07-13

**Rating:** 7
**Confidence:** 3
**Soundness:** 3 good
**Presentation:** 2 fair
**Contribution:** 4 excellent

**Summary:**

A more general variant of the predictive coding (PC) algorithm for learning deep neural networks is introduced. Unlike traditional PC, this generalized PC is also effective in training highly complex neural architectures such as transformer networks. Experiments are conducted on various tasks involving image generation and language modeling and similar performances to backpropagation-trained networks are obtained.

**Questions:**

- Why should one care about the different interpretation problem? In other words, in what sense is $PC_{\mathcal{F}}$ "inconsistent" and why is it important?
- Is there a generative model description of $PC_{\mathcal{F}\text{KL}}$ like that of $PC_{\mathcal{F}}$? How should the quantity $\ln p(\phi_l^{\mathcal{D}}|\mu_l^{\mathcal{D}})$ be interpreted?
- Is there a reason for the performance gap between BP and $PC_{\mathcal{F}\text{KL}}$ in figure 3 for $\mathcal{M}_3$?

**Limitations:**

The authors have adequately addressed the limitations of their work.

**Strengths And Weaknesses:**

Strengths:
- The contributions are very clearly stated.
- The method proposed is novel and it greatly generalizes the PC framework.
- The paper points out a concrete scenario for which traditional PC fails (i.e. the softmax activation) and clearly demonstrates the improvement of the newly proposed method by comparing different models with and without softmax. I find this to be convincing.
- The result of successfully training a transformer network with the newly proposed generalized PC is very impressive.

Weakness:
- The notation and exposition of the predictive coding methods can be quite confusing at times. For example on line 89: "[t]he nodes $x_l$ of each layer store the activation $\mu_l$," the usage of the word "store" can be misleading, making the reader question if $x_l$ is a variable/value or an abstract node object that "holds" values. There seems to be three different variables $\phi$, $\mu$ and $x$ that are closely related with each other, and the paper does not do a very good job of giving these variables concrete names/definitions. The distinction of different interpretations of the same value ($\cdot^{\mathcal{S}}$ vs. $\cdot^{\mathcal{D}}$) is also confusing in my opinion.
- On a similar note, figures 1 and 2 are also very difficult to read, since there are many different visual elements with different colors, line styles, sizes and labels, and some of them even overlap with each other. I would suggest making them simpler and/or adding more description for what each component represents.
- In the experiment section, no confindence intervals are shown for the training curves.

Overall the paper is well written with impressive results, although some notation and exposition can be improved.

---

> ### Author Response · Authors · 2022-08-02
> **Response to Reviewer Bynh (2/2)**
>
> ---
> > What is a possible interpretation of $\ln p(\phi^D|\mu^D)$?
>
> The general idea is that we can consider any layer as an independent probability distribution. The energy function is trying to balance between each two consecutive distributions. This happens accordingly to how $\ln p(\phi^D|\mu^D)$ is defined: a layer sends to its successor a series of samples sampled from its distribution, instead of a single one. In relation to the inconsistency problem, we can say that $\ln p(\phi^D|\mu^D)$ allows us to consider all the layers on the same “level”, being all distributions that communicate via samples. In the original formulation, instead, a layer was used directly as a sample and fed to the next one.
>
> ---
> > How can model $M_3$’s performance difference be explained?
>
> Note that all the experiments in that section (differently from the later experiments) are only a proof-of-concept of our method, and hence there was no independent hyperparameter search for the two experiments. Most likely, this is the reason why: the difference in performance can be due to the non-ideal hyperparameters and initialization of the weights in the predictive coding network, as we are also using the default Pytorch initialization, which was developed specifically for BP. (We did not perform any hyperparameter search here, as in this section the focus was more on showing the difference between PC and $PC_{F_{KL}}$, and the similarity between $PC_{F_{KL}}$ and BP in a randomly chosen architecture, by simply swapping in and out the training methods.) As it can be seen in the later experiments, $PC_{F_{KL}}$ and BP have very similar performance but require different hyperparameters and slightly different architectures (e.g., width and number of layers) to achieve the best results.

---

> ### Author Response · Authors · 2022-08-02
> **Response to Reviewer Bynh (1/2)**
>
> We thank the reviewer for the comments.
>
> ---
> > Could you please provide clarifications of the notations used?
>
> We added a table to summarise the notation and the usage of the different variables. See appendix F; the table will be integrated into the paper body. In particular, $x$, $\phi$, and $\mu$ are all linked to the value of the neurons in a neural network. $x$ is the generic term with which we refer to the neurons in a neural network. $\phi$ and $\mu$ represent the actual values assigned to the neurons $x$. This is the same notation used in the literature, e.g., by Millidge (2021). $\phi_{l}$ represents the value of the neurons of the PC layer $l$, while $\mu_l$ represents the activation value computed by the transformation present between layer $l$ and $l+1$ (i.e., $\mu_{l+1} = f_l(W_l \cdot \phi_l +b_l)$).  When using BP, we do not distinguish between $\phi$ and $\mu$ (since the value computed by a layer $l$ is directly fed to layer $l+1$), and therefore the neurons $x_l$ holds the value $\phi = \mu$ (with BP, the value of the neurons $x$ is implicitly stored in the computational graph, since the neuronal activations is necessary to compute the gradient in the sbackward pass). By clarifying this, the notations “$\cdot S$” and “$\cdot D$” should also become clearer. The notations $\cdot S$ and $\cdot D$ are there simply to show how differently a single value is “viewed” by the different PC formulations: in the first case, it is simply a vector of values, in the second, it represents the parameters of a probability distribution. It does not, in any case, imply any modification/transformation of the value itself.
>
> ---
> > The pictures are difficult to read, could you improve them?
>
> We addressed the problem by replacing figure 1 with a more explicit version, removing unnecessary elements and introducing a legend. This should also already clarify figure 2  See the updated Figure 1 in the revised paper. If the new version is better we will update figure 2 as well to match the new style.
>
> ---
> > Confidence intervals are missing in the experimental section. Is it possible to add them?
>
> We have added confidence intervals to the transformer’s test results, as it is the most impactful result. In general, the added interval does not change the conclusion reached in the paper.
>
> |    Method   | Test perplexity ($\mu \pm \sigma$) |
> |----------------|------------------------------|
> | BP             | 162.64 $\pm$ 0.76     |
> | $PC_{F_{KL}}$| 175.9081 $\pm$ 1.74 |
> | $PC$         | 590.08 $\pm$ 12.6     |
>
> ---
> > How is the original PC formulation  having different interpretations on $\phi$, why is it a problem and why is it important?
>
> We showed what we believe was the limiting factor in the original formulation of PC, namely, the inconsistent usage of the value $\phi$ of the neurons of a layer $l$, which is interpreted both as distribution parameters and as sample value. We showed how this distinction limits PC to Gaussian distributions, as it is the main scenario in which a vector can be interpreted both as a parameter (i.e., the mean of the distribution) and value. This also allows us to prove, in Appendix A, the equivalence of the new interpretation with the old one when only using Gaussian distributions.
>
> ---
> > Is there a generative model description for the new $PC_{F_{KL}}$?
>
> Yes, there is a generative model description of $PC_{F_{KL}}$. The formulation is actually analogous to the original $PC$. The first equality in Equation (4) still applies, we simply generalised what $p(x_l|x_{l-1})$ can represent. There is an alternative reformulation of PC that would result in the same energy function as $F_{KL}$, which would require changing the Direc-delta posterior assumption and result in a different generative formulation. Instead, by introducing the “KL via sampling” trick, we provide a constructive way to compute the energy within the same model and, at the same time, preserve the original generative formulation.
> For a more formal explanation, assuming as generative model
> $p(x_{0:L}) = p(x_0) \prod_{l=1}^L p(x_l|x_{l-1})$, (same as original one)
> and a posterior distribution
> $q_{\phi}(x_{0:L}) = \prod_{l=0}^L \delta(x_l-\phi_l)$, (same as original one)
> leads to the energy function
> $F = - E_{q_{\phi}(x_{0:L})}[\ln p(x_{0:L})]$. (same as original one)
> Here, we have a different expression of $\ln p(x_{0:L})$, having modified $p(x_l|x_{l-1})$. The specific generative model resulting from this expression depends on the particular distributions chosen for each layer.

---

### Meta-Review · Area_Chair_EzJp · 2022-08-25

**Recommendation:** Accept
**Confidence:** Less certain

**Metareview:**

Motivated by advancing the applicability of backpropagation alternatives, the paper extends predictive coding to non-Gaussian distributions, so it can be used to train effectively complex architectures such as transformers.

The reviews are divided: three reviews give a score of 7 (accept) whereas one review gives a score of 4 (borderline reject). The positive reviews cite the following strengths: clearly stated motivation, technical soundness, potential for impact and convincing experiments. The negative review cites lack of clarity as the main weakness (something also mentioned in one of the positive reviews), while the reviewer is not convinced about the originality of the method given similar advances in variational inference and generative modelling.

On balance, given the potential of impact in the field of predictive coding and the technical soundness, I'm happy to recommend acceptance, even though clarity is somewhat lacking.

Some reviewers requested more details on computational complexity, memory consumption and scalability. I encourage the authors to use the extra content page in the camera-ready version to discuss these aspects further.


**Award:**

No

---

### Decision · Program_Chairs · 2022-09-14

Accept

---

> ### Public Comment · ~Thomas_Tang1 · 2023-04-11
> **Very much confused with the math, could use some help**
>
> (1) In the paper, in equation (1), my believes is the conventional notation for KL divergence should be
> $\{D_{KL}}\left( {p\left( {x|o} \right)||q\left( {x|o} \right)} \right) = \sum\limits_x {p\left( {x|o} \right)} \\log \left( {\frac{{p\left( {x|o} \right)}}{{q\left( {x|o} \right)}}} \right) \ge 0\$
> instead, where the expectation is taken w.r.t. to the real intractable distribution $\{p\left( {x|o} \right)}\$. That also guarantees the KL divergence to be larger or equal to zero
>
> (2) In equation (2), not sure why $\ge$ and $\ln p(o)\$ are there, and isn't it $\\ln p(o) = \ln 1 = 0\$ ?
>
> (3) In equation (3), the expectation is shown to be w.r.t. the model/approximate distribution $\{q\left( {x|o} \right)}\$ and as mentioned in (1) to my believe it should not.
>
> As a result, since the first 3 equations seem to be off from my perspective which the rest of the paper is based on, I pretty much stuck there ...  Am I missing something? Could use some help here, thanks!

---

> > ### Public Comment · ~Luca_Pinchetti1 · 2023-04-11
> > **Reply to Thomas Tang**
> >
> > Hello!
> > Equations 1-6 are common in predictive coding literature. I suggest to look at here https://arxiv.org/abs/2107.12979 (chapter 2) to have a detailed step by step derivation of them. The notation is similar to the one used in the paper, so it should be easy to follow through.

---

> > > ### Public Comment · ~Thomas_Tang1 · 2023-04-11
> > > **Thank you for the quick response**
> > >
> > > Will check that out, thanks!